# Confounder-Free Continual Learning via Recursive Feature Normalization

**Yash Shah** [1]  **Camila Gonzalez** [1]  **Mohammad H. Abbasi** [1]  **Qingyu Zhao** [2]  **Kilian M. Pohl** [1]  **Ehsan Adeli** [1]

## Abstract

Confounders are extraneous variables that affect both the input and the target, resulting in spurious correlations and biased predictions. There are recent advances in dealing with or removing confounders in traditional models, such as metadata normalization (MDN), where the distribution of the learned features is adjusted based on the study confounders. However, in the context of continual learning, where a model learns continuously from new data over time without forgetting, learning feature representations that are invariant to confounders remains a significant challenge. To remove their influence from intermediate feature representations, we introduce the Recursive MDN (R-MDN) layer, which can be integrated into any deep learning architecture, including vision transformers, and at any model stage. R-MDN performs statistical regression via the recursive least squares algorithm to maintain and continually update an internal model *state* with respect to changing distributions of data and confounding variables. Our experiments demonstrate that R-MDN promotes equitable predictions across population groups, both within static learning and across different stages of continual learning, by reducing catastrophic forgetting caused by confounder effects changing over time.

## 1. Introduction

Confounders are study variables that influence both input and target, resulting in spurious correlations that distort the true underlying relationships within the data (Greenland & Morgenstern, 2001; Ferrari et al., 2020). These spurious correlations introduce bias into learning algorithms, causing the feature representations learned by deep neural networks (DNNs) to be skewed (Buolamwini & Gebru, 2018; Obermeyer et al., 2019; Oakden-Rayner et al., 2020; Chen et al., 2021; Seyyed-Kalantari et al., 2020).

This problem is particularly relevant in medical studies, such as those related to brain development (Casey et al., 2018), biological and behavioral health (Petersen et al., 2010; Brown et al., 2015), and dermatoscopic images (Tschandl et al., 2018), which are often confounded by demographic data such as age, sex, socio-economic background, and by factors like acquisition protocols and disease comorbidities. For example, a DNN trained to diagnose neurodegenerative disorders from brain MRIs could disproportionately rely on age instead of the underlying pathology. This may occur due to the disease causing accelerated aging or a selection bias, i.e., having different distributions in the diseased cohort versus the control group. This can lead to models that are inequitable and inaccurate for certain populations (Rao et al., 2017; Seyyed-Kalantari et al., 2020; Zhao et al., 2020; Adeli et al., 2020b; Lu et al., 2021; Vento et al., 2022). Given these challenges, it is crucial to develop techniques that enable DNNs to focus on task-relevant features while remaining invariant to confounders, which are often available as auxiliary information or metadata in the dataset.

Methods such as BR-Net (Adeli et al., 2020a), MDN (Lu et al., 2021), P-MDN (Vento et al., 2022), and RegBN (Ghahremani Boozandani & Wachinger, 2024) have previously been proposed to address the challenges posed by confounders when training DNNs. However, in continual learning, where data becomes available sequentially, traditional methods fail. This is because previous adversarial (BR-Net) or statistical (MDN, PMDN, etc.) methods need to use the entire dataset (implemented in batch-level statistics together with global memories) to estimate and remove the distribution of the features with respect to the confounders. In addition, because they require estimating batch-level statistics, they are unsuitable for modern architectures like vision transformers (Vaswani et al., 2017). A *continuum* of data may arise in various contexts. For example, in a cross-sectional study (Tschandl et al., 2018), the training process may be divided into distinct stages, each stage featuring different data distributions. In contrast, in a

---

[1]Stanford University, Stanford, United States [2]Weill Cornell Medicine, New York, United States. Correspondence to: Yash Shah <ynshah@stanford.edu>, Ehsan Adeli <eadeli@stanford.edu>.

*Proceedings of the 42nd International Conference on Machine Learning*, Vancouver, Canada. PMLR 267, 2025. Copyright 2025 by the author(s).

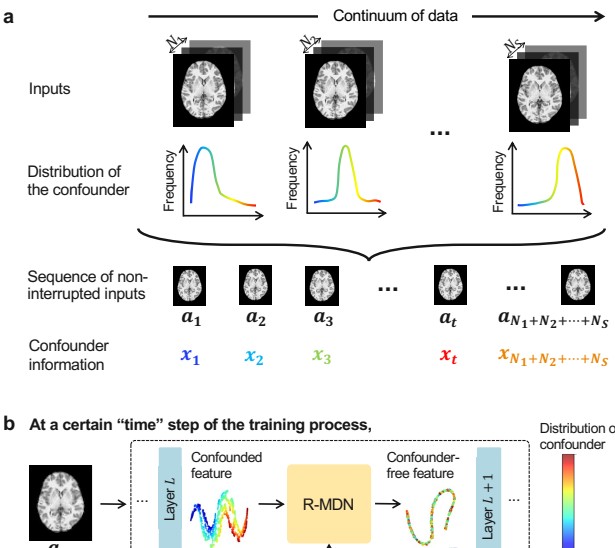

Figure 1. **Confounder-free continual representation learning. a.** A *continuum* of data with varying distributions of the confounder across different training stages continually passes through a DNN. **b.** An R-MDN layer removes the influence of confounders from the intermediate feature representations.

longitudinal study (Petersen et al., 2010; Brown et al., 2015; Casey et al., 2018), the system does not have access to all data at the outset; instead, new data—such as patient visits in a clinical study—continually arrive over an extended period, often spanning several years. These limitations of existing methods create a gap, as there is a need for algorithms that effectively and explicitly remove the influence of confounding variables under changing data distributions.

To this end, we propose *Recursive Metadata Normalization (R-MDN)* to remove (normalize) the effects of the confounding variables from the learned features of a DNN through statistical regression. Specifically, R-MDN leverages the recursive least squares (RLS) algorithm (Albert & Sittler, 1965), which has been widely used in adaptive filtering, control systems for reinforcement learning, and online learning scenarios (Xu et al., 2002; Gao et al., 2020). R-MDN is a layer that *can be inserted at any stage within a DNN*. The use of statistical linear regression is motivated by its success in de-confounding learned feature representations (Mc-Namee, 2005; Brookhart et al., 2010; Pourhoseingholi et al., 2012; Adeli et al., 2018). Similar to the original MDN formulation, R-MDN does not assume that confounders have to linearly influence the input image because the method does not directly operate on images, but rather on the feature embeddings, which are non-linear abstractions of the input. Moreover, R-MDN can be applied to feature embeddings at multiple layers, so overall it can effectively remove non-linear confounding between the input and the confounders. R-MDN operates by iteratively updating its in-

ternal parameters—consisting of regression coefficients and an estimated inverse covariance matrix, which together form an internal model *state*—based on previously computed values whenever new data are received. This state represents the current understanding of the relationship between the learned features and the confounders, enabling the model to adapt dynamically as new data flows in.

R-MDN, therefore, applies to continual learning, where each training stage consists of data drawn from different stationary distributions. This *continuum* of data can be understood as a sequence of uninterrupted examples that a model learns from *over time*. Here, R-MDN does not need to train a stage-specific network. Instead, the internal state can be continuously updated over time as the model progresses through successive training stages. As a result, only a single network equipped with R-MDN layers needs to be trained on the entire dataset, with the model being able to generalize across stages (data or confounder distributions)—maintaining stable performance while also removing the effects of the confounders (see Figure 1).

In summary, we propose R-MDN—a flexible normalization layer that is able to residualize the effects of confounding variables from learned feature representations of a DNN by leveraging the recursive least squares closed-form solution. It can do so under varying data or confounder distributions, making it an effective algorithm for continual learning. We provide a theoretical foundation for our approach (Section 3) and empirically validate it in different experimental setups (Sections 4.1, 4.2, 4.3). Furthermore, we demonstrate the applicability of R-MDN to static learning (Section C, 4.4) and architectures that prohibit the calculation of batch-level statistics (Sections 4.2, 4.3). We find that R-MDN helps to make equitable predictions for population groups not only within a single cross-sectional study (Section 4.4), but also across different stages of training during continual learning (Sections 4.1, 4.2, 4.3), by minimizing catastrophic forgetting of confounder effects over time.[1]

## 2. Related Work

Widely used techniques such as batch (Ioffe, 2015), layer (Ba et al., 2016), instance (Ulyanov, 2016), and group (Wu & He, 2018) normalization standardize intermediate feature representations of DNNs, i.e., they normalize them to have zero mean and unit standard deviation across different representational axes. They do not explicitly remove the effects of confounding variables from these features.

Prior works have proposed methods for learning confounder-invariant feature representations based on domain-adversarial training (Liu et al., 2018; Wang et al., 2018;

---

[1]The implementation code is available at https://github.com/stanfordtailab/RMDN.git.

Sadeghi et al., 2019; Adeli et al., 2020a), closed-form statistical linear regression analysis (Lu et al., 2021), penalty approach to gradient descent (Vento et al., 2022), regularization (Ghahremani Boozandani & Wachinger, 2024), disentanglement (Liu et al., 2021; Tartaglione et al., 2021), counterfactual generative modeling (Neto, 2020; Lahiri et al., 2022), fair inference (Baharlouei et al., 2020), and distribution matching (Baktashmotlagh et al., 2016; Cao et al., 2018). Among these, distribution matching techniques do not specifically remove the influence of individual confounders from learned features. Adversarial training, on the other hand, typically involves a confounder-prediction network applied to pre-logits feature representations, with an adversarial loss used to minimize the correlation between features and confounders. However, adversarial approaches struggle to scale effectively when faced with multiple confounding variables. Likewise, disentanglement, fair inference, and counterfactual generative modeling techniques only partially remove confounder effects from a single layer of the network (Zhao et al., 2020; Vento et al., 2022).

Among the methods listed earlier, Metadata Normalization (MDN) (Lu et al., 2021), which uses statistical regression analysis, is a popular technique. MDN is a layer that can be inserted into the DNN to residualize confounder effects from intermediate learned features. It does so through the ordinary least squares algorithm, wherein it computes a closed form solution for the expression $z = X\beta + r$ as $\beta = \left(X^\top X\right)^{-1} X^\top z$, where $z$ is the intermediate learned feature vector, $X$ is the confounder matrix, $\beta$ are regression coefficients, and $r$ is the component in the learned features invariant to the confounder. To work with minibatches of data, MDN re-formulates the closed-form solution as $\beta = N\Sigma^{-1}\mathbb{E}(xz)$, where $\Sigma^{-1} = \left(X^\top X\right)^{-1}$ is pre-computed with respect to training data, with the amount being how much is needed to estimate its distribution (ideally, the entire dataset) at the start of training, and the expectation $\mathbb{E}(xz)$ is computed using batch-level estimates during training. In the context of continual learning where we do not have all data at the outset of training, or where an infinite replay buffer is not assumed, this pre-computation step prohibits effective transfer of performance to training stages with different data distributions. Even in static learning, employing batch-level statistics during training precludes MDN from being used with architectures such as vision transformers, where computation is parallelized over individual examples.

To alleviate issues around the use of batch statistics, a penalty approach to MDN (P-MDN) was proposed (Vento et al., 2022). The authors of P-MDN observe that MDN solves a bi-level nested optimization problem by having the network learn task-relevant features while also being invariant to the confounder, and instead suggest to solve a proxy objective $\min_{\beta,W} \mathcal{L}(\varphi(z - X\beta), y) + \gamma\mathcal{L}^*(z; X)$, where $\varphi$ is the non-linear computation to be performed within the

network after the current layer, $y$ are the target labels, and $\gamma$ is a penalty parameter that trades off task learning with confounder-free feature learning. After these modifications, P-MDN is able to work with arbitrary batch sizes. However, as we see in the results of this work, $\gamma$ becomes very difficult to tune, and optimizing the proxy objective often leads to non-robust results with high variance across different runs.

For continual learning, methods based on regularization (Kirkpatrick et al., 2017), knowledge distillation (Li & Hoiem, 2017), and architectural changes (Rusu et al., 2016; Mallya & Lazebnik, 2018; Bayasi et al., 2024) have been proposed to overcome *catastrophic forgetting*—the phenomenon where DNNs forget information learned in prior training stages when acquiring new knowledge. Some of these methods are motivated by dealing with task (domain) specific biases by learning task (domain) general features (Arjovsky et al., 2019; Zhao et al., 2019; Creager et al., 2021). Such approaches, however, do not leverage metadata to remove effects due to specific confounders from the learned features. While domain-adversarial training and P-MDN still apply to the continual learning setting, we show in this paper that they do not perform well in many scenarios.

## 3. Methodology

Consider that we have $N$ training samples, where the input matrix $A \in \mathbb{R}^{N \times d}$ for some dimension $d$, is associated with target labels $y \in \mathbb{R}^N$ and information about the confounding variable $\tilde{x} \in \mathbb{R}^N$. Let the output after a particular layer of a deep network be the features $z \in \mathbb{R}^N$. Our goal is to obtain the residual $r$ from the expression $z = \tilde{x}\tilde{\beta}_x + y\tilde{\beta}_y + r = X\beta + r$, where $X = [\tilde{x}\ y]$ and $\beta = [\tilde{\beta}_x; \tilde{\beta}_y]$ is a set of learnable parameters. In other words, the learned features $z$ are first projected onto the subspace spanned by the confounding variable and the labels, with the term $\tilde{x}\tilde{\beta}_x$ corresponding to the component in $z$ explained by the confounder, and $y\tilde{\beta}_y$ to that explained by the labels. We want to remove the influence of $\tilde{x}$ from $z$ while preserving the variance related to the labels. We thus compute the composite $\beta$ as theorized below, but obtain the residual $r = z - \tilde{x}\tilde{\beta}_x$; i.e., only with respect to $\tilde{\beta}_x$. This residual explains the components in the intermediate features that are irrelevant to the confounder but relevant to the labels and, thus, for the classification task.

To accomplish this, we use the recursive least squares approach. We start from the closed-form solution found for the ordinary least squares (OLS) estimator:

$$\beta = \left(\sum_{i=1}^N X_{i,:} X_{i,:}^\top\right)^{-1} \left(\sum_{i=1}^N z_i X_{i,:}\right), \qquad (1)$$

where $X_{i,:}$ is the $i^\text{th}$ row of $X$. If we represent $R(N) =$

$\sum_{i=1}^{N} X_{i,:} X_{i,:}^{\top}$ and $Q(N) = \sum_{i=1}^{N} z_i X_{i,:}$, this is equivalent to writing $\beta = R(N)^{-1} Q(N)$.

Now, say that we have a new sample $A_{N+1,:}$ come in. The confounding variable and intermediate features for this sample are $X_{N+1,:}$ and $z_{N+1}$ respectively. This means that we need to compute new parameters:

$$\beta' = R(N+1)^{-1} Q(N+1) \tag{2}$$
$$= \left(R(N) + X_{N+1,:} X_{N+1,:}^{\top}\right)^{-1} \left(Q(N) + z_{N+1} X_{N+1,:}\right)$$

Fortunately, $R(N+1)^{-1}$ can be efficiently computed using the Sherman-Morrison rank-1 update rule (Sherman & Morrison, 1950):

$$\left(R(N) + X_{N+1,:} X_{N+1,:}^{\top}\right)^{-1}$$
$$= R(N)^{-1} - \frac{R(N)^{-1} X_{N+1,:} X_{N+1,:}^{\top} R(N)^{-1}}{1 + X_{N+1,:}^{\top} R(N)^{-1} X_{N+1,:}}. \tag{3}$$

With training, $Q(N)$ changes as the features $z_i$ learned by the model change. In a continual learning setting, we cannot recompute $Q(N)$ and instead take it to be an estimate of the true value. Empirically, we see that this is a reliable estimate. The entire derivation is presented in Suppl. A. Importantly, applying R-MDN does not incur significant computational and memory overhead (a detailed analysis is presented in Suppl. B).

## 3.1. Estimates produced by R-MDN

Stoica & Åhgren (2002) theorize that RLS (i.e., the linear model constructed by R-MDN) provides an estimate that coincides with the estimate from OLS (used by MDN) as the amount of data $N \to \infty$. However, a careful initialization of the inverse covariance matrix can speed up this convergence. We thus initialize $R(0)^{-1} = \epsilon I$, where $\epsilon > 0$ is a small scalar, as most commonly used by prior works (Haykin, 2002; Stoica & Åhgren, 2002; Liu et al., 2009; Skretting &

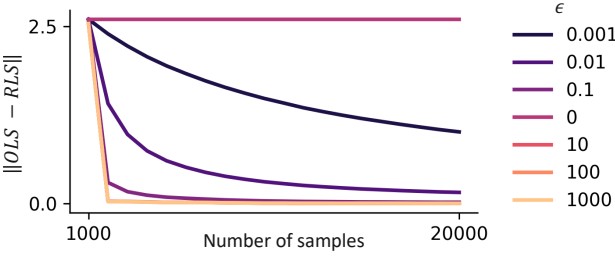

*Figure 2.* $\ell_2$ norm between estimates produced from the linear model constructed by R-MDN and MDN on synthetically generated data when varying $\epsilon$. Estimates converge fairly quickly and for a range of hyperparameter initializations.

Engan, 2010). From Figure 2, for a range of $\epsilon$ values such as 10, 100, and 1000, the RLS estimates approach that of OLS quite quickly.

## 3.2. Mini-batch learning

So far, we have theorized R-MDN within an online learning setting, with the system adapting to new information as it comes in one at a time. However, we can extrapolate this method to work with mini-batches of data. Consider that the system receives new mini-batches of information $\hat{X} \in \mathbb{R}^{B \times d}$, for some batch size $B$. We can adapt the R-MDN approach by now using the Sherman-Morrison-Woodbury formula (Woodbury, 1950):

$$\left(P + \hat{X}\hat{X}^{\top}\right)^{-1}$$
$$= P^{-1} - P^{-1}\hat{X}\left(I + \hat{X}^{\top}P^{-1}\hat{X}\right)^{-1}\hat{X}^{\top}P^{-1}. \tag{4}$$

## 3.3. From batch to layer statistics

Remember that one of the drawbacks of MDN is that it has to compute and store batch-level statistics $\Sigma$ with respect to the entire training data prior to training (refer to Section 2, 3rd paragraph). This requirement makes MDN unsuitable for modern architectures like vision transformers, wherein computations happen in parallel over all examples in a mini-batch. Incorporating an MDN module will inherently require an *aggregation* step for the computation of batch-level statistics, resulting in significant computational overhead. R-MDN, on the other hand, operates on the level of individual examples in a minibatch. Thus, it works in a purely online regime and can be inserted in vision transformers to residualize intermediate learned features.

## 3.4. Regularization

R-MDN can adapt quickly to changing data distributions over time due to its iterative nature. However, this iterative nature of the method might sometimes make it too sensitive to small changes in the data. Random fluctuations, or data noise, can lead to unstable updates to R-MDN parameters. Therefore, we add a regularization term $\lambda I$ to $P(N+B)$. $\lambda$ is a hyperparameter that is tuned during training (ablation in Suppl. H). This has the effect of smoothing out the updates and stabilizing the residualization process, resulting in some robustness to noise. Additionally, it helps to ensure numerical stability by preventing the computation of an inverse for a matrix that might be singular or ill-conditioned.

# 4. Experimental Results

We train on a *continuum* of data by slightly modifying the setting described by Lopez-Paz & Ranzato (2017): A 4-tuple $(a_i, x_i, y_i, s_i)$ for $i \in [N]$, where $a_i \in A$ is the in-

*Table 1.* **Quantifying metrics for the synthetic dataset in continual learning.** ACCd, BWTd ($\times 10^{-2}$), and FWTd mean and standard deviation for different methods and datasets. A total of 5 runs were performed with different model initialization seeds. We compare against prior works—BR-Net (Adeli et al., 2020a), MDN (Lu et al., 2021), and P-MDN (Vento et al., 2022). MDN, a CNN was trained separately on each training stage and then evaluated against all stages. Best results in **bold**, second-to-best, underlined.

| Method | Dataset 1 Confounder dist. changes | | | Dataset 2 Main effects change | | | Dataset 3 Both distributions change | | |
|---|---|---|---|---|---|---|---|---|---|
| | ACCd | BWTd | FWTd | ACCd | BWTd | FWTd | ACCd | BWTd | FWTd |
| CNN Baseline | $0.18 \pm 0.00$ | **$0.03 \pm 0.04$** | $0.19 \pm 0.00$ | $0.2 \pm 0.00$ | **$-0.05 \pm 0.13$** | $0.21 \pm 0.00$ | $0.28 \pm 0.00$ | $\underline{-0.37 \pm 0.02}$ | $0.31 \pm 0.00$ |
| BR-Net | $0.04 \pm 0.03$ | $-1.28 \pm 1.37$ | $0.05 \pm 0.04$ | $0.04 \pm 0.02$ | $-1.43 \pm 1.81$ | $0.04 \pm 0.03$ | $0.07 \pm 0.04$ | $-0.55 \pm 0.56$ | $0.08 \pm 0.04$ |
| Stage-specific MDN | $0.25 \pm 0.00$ | $16.2 \pm 2.27$ | $0.09 \pm 0.02$ | $0.13 \pm 0.00$ | $12.2 \pm 0.57$ | **$0.02 \pm 0.01$** | $0.13 \pm 0.00$ | $11.7 \pm 1.48$ | $\underline{0.02 \pm 0.01}$ |
| P-MDN | $\underline{0.04 \pm 0.01}$ | $-0.61 \pm 1.37$ | $\underline{0.04 \pm 0.01}$ | $\underline{0.04 \pm 0.00}$ | $-0.76 \pm 2.46$ | $0.05 \pm 0.01$ | $\underline{0.05 \pm 0.01}$ | $-1.37 \pm 2.21$ | $0.06 \pm 0.01$ |
| R-MDN | **$0.02 \pm 0.01$** | $\underline{0.15 \pm 0.14}$ | **$0.02 \pm 0.01$** | **$0.03 \pm 0.00$** | $\underline{0.07 \pm 2.17}$ | $\underline{0.03 \pm 0.00}$ | **$0.02 \pm 0.01$** | **$-0.04 \pm 0.78$** | **$0.02 \pm 0.00$** |

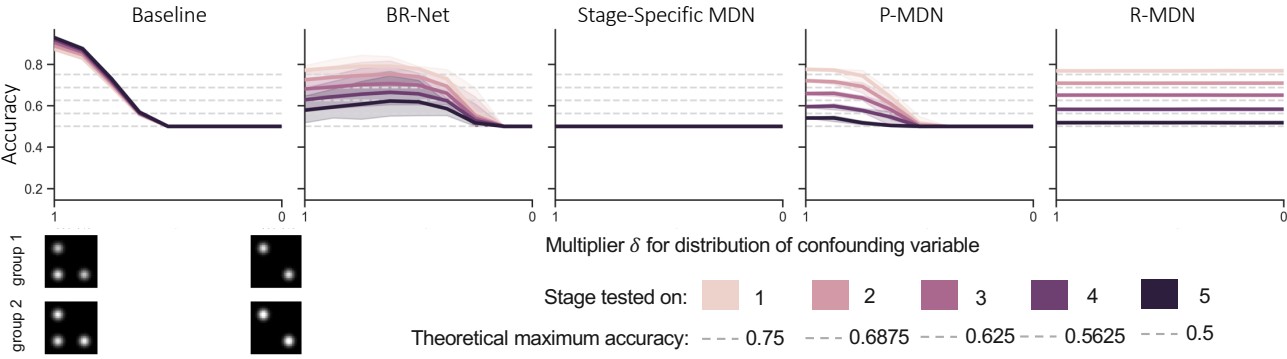

*Figure 3.* **Effects of the presence of the confounder for task generalization.** After training the different methods on data with confounders, we evaluate them on test data with varying intensities of the confounder, from 1 (completely present) to 0 (completely absent). R-MDN does not get influenced by the change in confounder distribution during inference as it maintains its prediction accuracy (i.e., the lines are straight). This shows that it does not "cheat" by looking at the confounder and instead learns task-relevant features.

put, $x_i \in \boldsymbol{X}$ is the confounder, $y_i \in \boldsymbol{Y}$ is the label, and $s_i \in \boldsymbol{S}$ is the training stage descriptor, satisfies *local iid*, i.e., $(a_i, x_i, y_i) \overset{\text{iid}}{\sim} \mathcal{P}_{s_i}(\boldsymbol{A}, \boldsymbol{X}, \boldsymbol{Y})$. Our goal is to learn a classifier $g : \boldsymbol{A} \times \boldsymbol{S} \to \boldsymbol{Y}$, that is able to predict the label $y$ associated with an unseen 2-tuple $(a, s)$, where $(a, y) \sim \mathcal{P}_s$ *at any point* during or after training on the $\boldsymbol{S}$ stages, in a way such that it does not use confounder information in $x$.

### 4.1. A Continuum of Synthetic Datasets

We first use a synthetic dataset with varying distributions of the confounding variable and main effects as a playground to test how models behave under controlled variations in individual variables. Specifically, we design 3 different datasets, each with 5 stages of training that arrive sequentially. Over time, the distributions of the confounder, the main effects, or both change in a way that emphasizes biased learning of a classifier that uses confounder information for discrimination. A complete description is presented in Suppl. E.

To quantify knowledge transfer, we define the Average Accuracy distance (ACCd), Backward Transfer distance (BWTd),

and Forward Transfer distance (FWTd) metrics. These metrics are adapted from ACC, BWT, and FWT defined by Lopez-Paz & Ranzato (2017) for the setting where a model is expected to achieve certain theoretical accuracies on data from both past and future stages of training. Say we have a total of $S$ stages. Let $R_{i,j}$ denote the classification accuracy of the model on stage $s_j$ after learning stage $s_i$. And let $\boldsymbol{A}_i$ denote the theoretical maximum accuracy for stage $s_i$. Then,

$$\text{ACCd} = \frac{1}{S} \sum_{i=1}^{S} |R_{S,i} - \boldsymbol{A}_i|, \quad (5)$$

$$\text{BWTd} = \frac{1}{S-1} \sum_{i=1}^{S-1} |R_{S,i} - \boldsymbol{A}_i| - |R_{i,i} - \boldsymbol{A}_i|, \quad (6)$$

$$\text{FWTd} = \frac{1}{S-1} \sum_{i=2}^{S} |R_{i-1,i} - \boldsymbol{A}_i|. \quad (7)$$

The smaller the metrics, the better the model. We use a con-

*Table 2.* **Training a 2D vision transformer on synthetic continual data.** The setup is the same as in Section 4.1. We use dataset 3—where the distributions of both the confounder and main effects change. Results are averaged over 5 runs of random model initialization seeds.

| Method | ACCd | BWTd | FWTd |
|---|---|---|---|
| ViT Baseline | 0.110 ± 0.027 | -0.173 ± 0.024 | 0.306 ± 0.009 |
| R-MDN | **0.061 ± 0.013** | **-0.111 ± 0.019** | **0.188 ± 0.015** |

volutional neural network backbone for each method (details in Suppl. F). A stage-specific MDN does not perform well on the average accuracy metric, as indicated by a high ACCd (Table 1 and Figure 12b). While other methods are good at backward transfer, R-MDN also improves forward transfer. This means that even with changing distributions of the confounding variable, R-MDN only "looks at" the main effects for classification, learning features that transfer well to later tasks while remaining invariant to the confounder itself. Other methods make use of confounders to various degrees, pulling their classification accuracy away from $A$. This is also reflected in R-MDN consistently achieving an accuracy close to the theoretical maximum for the test sets of each stage, while also showing the lowest correlation with the confounding variable (Figure 12c). This outcome is also reflected when starting with a vision transformer backbone (Table 2), and when constructing a slightly more complex synthetic setup where we vary two different axes of possibly confounding information: position and intensity of variables (Suppl. D).

We further quantify how different methods generalize to unseen images where the confounder is absent (Figure 3, Suppl. K). This issue arises in situations where, for instance, a model is trained on data from multiple hospitals and then tested on data from a previously-unseen one (Zech et al., 2018). In this scenario, the base model experiences a sharp drop in performance when the distribution of the confounder changes in the test data. Both BR-Net and P-MDN show

*Table 3.* **Comparing R-MDN against recent analytic continual learning frameworks.** We compare R-MDN against ACIL (Zhuang et al., 2022) and F-OAL (Zhuang et al., 2024) under a CNN backbone. The setup is the same as in Section 4.1. We use dataset 3—where the distributions of both the confounder and main effects change. Results are averaged over 3 runs of random model initialization seeds.

| Method | ACCd | BWTd | FWTd |
|---|---|---|---|
| ACIL | 0.29 ± 0.00 | **-0.00 ± 0.00** | 0.30 ± 0.00 |
| F-OAL | 0.28 ± 0.00 | **0.01 ± 0.00** | 0.28 ± 0.00 |
| R-MDN | **0.02 ± 0.01** | **-0.00 ± 0.01** | **0.02 ± 0.00** |

some resistance to the distribution shift but fail when the confounder is entirely absent. In contrast, R-MDN maintains consistent performance across all distributions.

In addition to comparing with prior methods such as MDN, BR-Net, and P-MDN that have been proposed to remove the influence of confounders from learned features, we also want to compare against recently proposed methods in the continual learning setting. Analytic continual learning methods such as ACIL (Zhuang et al., 2022) and F-OAL (Zhuang et al., 2024) share some similarities to our method in terms of using recursive least squares, although they address an inherently different problem that regardless is interesting to compare against. These methods address catastrophic forgetting and improve task accuracy, but do not explicitly remove the influence of confounders from learned features. R-MDN is a normalization layer, and can be integrated with various continual learning frameworks, as we show later in Section 4.2. In Table 3, we see that both ACIL and F-OAL have excellent BWTd, which means that they effectively mitigate catastrophic forgetting, as their papers propose. However, they result in significantly worse ACCd and FWTd, which means that they make use of confounder information to make predictions (exhibiting large deviations from the theoretical maximum accuracy). On the other hand, R-MDN has better BWTd, ACCd, and FWTd, meaning that it learns confounder-free features for making predictions, thus approaching the theoretical accuracy.

### 4.2. HAM10000 Skin Lesion Classification

Next, we classify 2D dermatoscopic images of pigmented skin lesions into seven distinct diagnostic categories with the HAM10000 dataset (Tschandl et al., 2018). The dataset consists of 10015 images, which we divide into five training stages. In each stage, the age distribution—the confounding variable (see Suppl. E)—varies, with younger populations represented in the earlier stages and older populations in the later stages. For each stage, we randomly allocate 80% of the images for training and the remaining 20% for testing.

In this experiment, we utilize a vision transformer as the base architecture and as the encoder for BR-Net. For R-MDN, we explore three different variants: (A) inserting the R-MDN layer after the self-attention layer in every transformer block, as well as after the pre-logits layer; (B) inserting it at the end of every transformer block and after the pre-logits layer; and (C) inserting it only after the pre-logits layer. For P-MDN, we place the P-MDN layer right after the pre-logits layer. Additionally, we compare to three continual learning methods as baselines: elastic weight consolidation (EWC) (Kirkpatrick et al., 2017) as a regularization method, learning without forgetting (LwF) (Li & Hoiem, 2017) for knowledge distillation, and PackNet (Mallya & Lazebnik, 2018), an architectural method that applies iterative pruning.

*Table 4.* **HAM10K skin lesion classification results for continual learning.** Results shown as the mean and standard deviation over test sets of different stages of training for the model after being trained on the last training stage. Best and second-to-best results shown in bold and underlined respectively. Our metrics of interest are dcor$^2$, BWT, and FWT. A higher accuracy in our case does not equate to a better model as it might be "cheating" by looking at the confounder for prediction.

| Method | Accuracy | Average dcor$^2$ | BWT | FWT |
|---|---|---|---|---|
| ViT Baseline | $0.7095 \pm 0.0626$ | $0.0864 \pm 0.0336$ | $0.0278 \pm 0.0446$ | $\underline{0.5125 \pm 0.0705}$ |
| BR-Net (Adeli et al., 2020a) | $0.7247 \pm 0.0627$ | $\underline{0.0544 \pm 0.0534}$ | $-0.0207 \pm 0.0166$ | $\mathbf{0.5592 \pm 0.0897}$ |
| P-MDN (Vento et al., 2022) | $0.6750 \pm 0.0945$ | $0.2595 \pm 0.0620$ | $\underline{0.0706 \pm 0.0622}$ | $0.4391 \pm 0.0372$ |
| R-MDN (A) | $0.5503 \pm 0.0541$ | $0.0928 \pm 0.0630$ | $-0.0268 \pm 0.0248$ | $0.4130 \pm 0.0709$ |
| R-MDN (B) | $0.5288 \pm 0.0571$ | $0.0739 \pm 0.0555$ | $0.0571 \pm 0.0693$ | $0.3362 \pm 0.0881$ |
| R-MDN (C) | $0.6919 \pm 0.0723$ | $\mathbf{0.0475 \pm 0.0247}$ | $\mathbf{0.1246 \pm 0.2123}$ | $0.3997 \pm 0.1555$ |
| EWC (Kirkpatrick et al., 2017) | $0.6437 \pm 0.0586$ | $0.0938 \pm 0.0506$ | $0.0698 \pm 0.0238$ | $0.4457 \pm 0.0620$ |
| EWC + R-MDN (C) | $0.6739 \pm 0.0686$ | $0.0592 \pm 0.0488$ | $0.0754 \pm 0.1614$ | $0.4404 \pm 0.1305$ |
| LwF (Li & Hoiem, 2017) | $0.7356 \pm 0.0757$ | $0.0512 \pm 0.0407$ | $0.0387 \pm 0.0390$ | $0.5277 \pm 0.0605$ |
| LwF + R-MDN (C) | $0.7186 \pm 0.0736$ | $0.0354 \pm 0.0210$ | $0.1348 \pm 0.1994$ | $0.4434 \pm 0.1403$ |
| PackNet (Mallya & Lazebnik, 2018) | $0.6849 \pm 0.0745$ | $0.0470 \pm 0.0304$ | $0.0538 \pm 0.0670$ | $0.4965 \pm 0.0611$ |

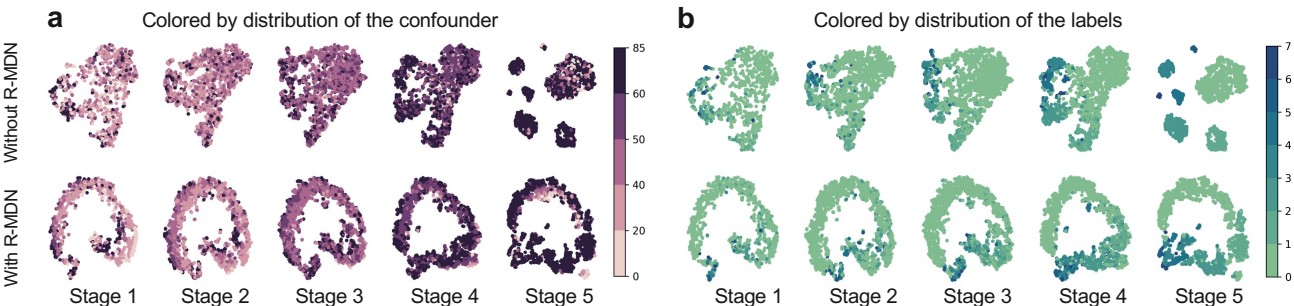

*Figure 4.* **Visualizing learned features for HAM10K skin lesion classification.** t-SNE representation of the learned features after training a model with and without R-MDN on all stages of continual learning. Without R-MDN, the model drives fine-grained separation of feature clusters based on category labels, especially for stage 5, which roughly overlap with confounder distributions for that training stage. By looking at the confounder for fine-grained discrimination, the baseline fails to generalize to previous data. R-MDN, on the other hand, does not seem to be using confounder information to drive fine-grained feature separation, leading to better backward transfer.

Results are summarized in Table 4 and visualized in Figure 14. While the base model performs decently on the classification task, it exhibits significantly lower backward transfer in earlier stages of training. In contrast, R-MDN not only effectively removes the confounder's influence from the features, as indicated by a low *dcor*$^2$ value, but also demonstrates significantly better backward transfer than the base model. We explore this effect by observing the t-SNE visualizations of their features (Figure 4). When the base model is trained in the final stage, it learns to clearly separate feature clusters from each of the seven diagnostic categories, especially after being trained on stage 5 (Figure 4b). However, it is possible that this separation is influenced by the stage-specific distribution of the confounding variable (Figure 4a), leading to spurious correlations driving cluster separation and, thus, poor transfer to previous data. On the other hand, R-MDN—which relies on task-relevant information—also forms feature clusters for the different

categories but without introducing the same level of separation possible by looking at the confounder. R-MDN is able to apply the knowledge learned in the current stage to previous stages of training, improving its overall backward transfer performance.

Of the three R-MDN variants, R-MDN (C) performs the best. Moreover, applying R-MDN to classic continual learning frameworks such as EWC and LwF still drives the correlation with the confounder significantly down. In contrast, other methods, such as BR-Net and P-MDN, do not perform as well. BR-Net catastrophically forgets past information, and P-MDN fails to effectively remove confounder effects.

### 4.3. ADNI Diagnostic Classification

Finally, we move from a cross-sectional dataset to a longitudinal study. The Alzheimer's Disease Neuroimaging Initiative is a multi-center observational study that collects

*Table 5.* **ADNI diagnostic classification for continual learning.** Average squared distance correlation after training on the last training stage for different groups across different methods (Baseline ViT, P-MDN (Vento et al., 2022), and R-MDN). Note that we do not train BR-Net as adversarial training does not scale well to more than one confounder (Zhao et al., 2020).

| Method | $dcor^2$ (MCI) | $dcor^2$ (CN) | $dcor^2$ (AD) |
|--------|----------------|---------------|---------------|
| ViT Baseline | $0.20 \pm 0.09$ | $0.20 \pm 0.14$ | $0.25 \pm 0.22$ |
| P-MDN | $\mathbf{0.06 \pm 0.05}$ | $\underline{0.08 \pm 0.05}$ | $\underline{0.24 \pm 0.16}$ |
| R-MDN | $\underline{0.08 \pm 0.06}$ | $\mathbf{0.05 \pm 0.04}$ | $\mathbf{0.09 \pm 0.07}$ |

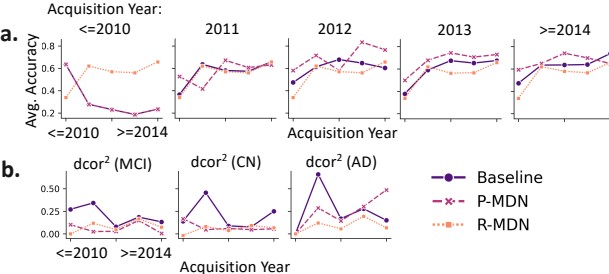

*Figure 5.* **Visualizations for ADNI diagnostic classification. a.** Average accuracy and **b.** squared distance correlation for each participant group for different stages of training. The baseline and P-MDN have a poor forward transfer for the first acquisition year, as they use confounder information to overfit to that training stage. R-MDN has a significantly better performance on future training stages as it does not look at the confounder, as indicated by a lower $dcor^2$ for all diagnostic groups.

neuroimaging data from participants that fall into different diagnostic groups over several years (Mueller et al., 2005; Petersen et al., 2010). We pose a diagnostic classification task—Alzheimer's Disease (AD), Mild Cognitive Impairment (MCI), and Control (CN)— from T1w MRIs under changing distributions of age and sex, the two confounders for the experiment (additional details provided in Suppl. E). We train a 3D vision transformer as the base architecture.

Similar to previous experiments, applying R-MDN results in a lower squared distance correlation, implying that it does not take into account confounders when making predictions (Table 5, Figure 5b). P-MDN does the same only to some extent, as it is not able to drive correlation down for all three participant groups. This leads to the baseline and P-MDN models "cheating" by using discriminatory cues from the confounders to better classify inputs and demonstrating a slightly higher average accuracy (Figure 5a). This occurs at the expense of not being able to generalize to future stages of training, as, for example, seen by a roughly monotonically decreasing accuracy curve for the acquisition year $\leq$2010 (Figure 5a). R-MDN learns task-relevant cues, characterizing model performance in the absence of "cheating".

## 4.4. Static Learning

R-MDN is not only helpful in continual settings, but also an effective regularizer for static training. To demonstrate this, we explore a setting where the system only receives data from a single stationary distribution. Experiments were conducted on a proof-of-principle synthetic dataset (Suppl. C) and a neuroimaging dataset.

**ABCD Sex Classification:** We used T1w MRIs from the ABCD (Adolescent Brain Cognitive Development) study (Casey et al., 2018) for the task of binary sex classification. Within a cross-sectional study setting, we take 10686 baseline (i.e., first visit) MRIs, confounded by scores from the Pubertal Development Scale (PDS)—a validated measure of pubertal stage identified through self-assessment. PDS is a confounder because it is significantly (Suppl. E) larger in girls ($2.182 \pm 0.9$) than in boys ($1.372 \pm 0.6$) (Adeli et al., 2020b). PDS categorizes participants as either (1) pre-pubertal, (2) early-pubertal, (3) mid-pubertal, (4) late-pubertal, or (5) post-pubertal (Carskadon & Acebo, 1993).

We start with a 3D CNN as the base architecture consisting of three stacks of convolutional layers, each followed by ReLU non-linearity and max pooling, and ending with two fully connected layers. We insert a residualization module after every layer except the last one. In addition to this approach, we establish two additional baselines—one where we use BR-Net, and adversarial training framework, with the same base model as the encoder, and another where we pre-process the input images prior to training by regressing out the influence of confounders directly from the pixel space (hereafter referred to as *Pixel-Space MDN*). We set the batch size to 128, which is the largest that can be fit in GPU memory (see additional details in Suppl. F).

We observe that the base model has high accuracy, but at the cost of being significantly biased towards girls—its feature representations have a larger $dcor^2$ with the confounder for girls than boys, and a higher true negative rate (Table 6). This is because the base model makes use of the pubertal development to drive its predictions. On the other hand, R-MDN incurs a modest decrease in performance but significantly drives down the correlation between the learned features and the confounder for both boys and girls. Moreover, it has the lowest mean difference between the true positive and true negative rates among all evaluated methods (quantified in Table 6, visualized in Figure 13a,b,c), signifying that it is not biased toward children of either sex. Other methods such as MDN and P-MDN have a higher prediction accuracy but either fail to drive down the correlation between the features and the confounder due to requiring relatively larger batch sizes or remain biased towards girls despite driving the correlation down. As the authors of the MDN paper hypothesize, batch-level statistics could be aggregated over several batches to virtually increase the batch

*Table 6.* **ABCD sex classification results for static learning.** Accuracy, true positive (TPR) and negative rates (TNR), difference between TPR and TNR, and squared distance correlation for both boys and girls for different methods. Results are shown as the mean and standard deviation over 5 folds of 5-fold cross-validation, with data split by subject and site ID. Our metrics of interest are (TPR - TNR) and dcor$^2$. A higher accuracy in our case does not equate to a better model as it might be "cheating" by looking at the confounder for prediction.

| Method | Accuracy | TPR | TNR | TPR - TNR | dcor$^2$ (boys) | dcor$^2$ (girls) |
|---|---|---|---|---|---|---|
| CNN Baseline | 86.86 ± 0.354 | 85.41 ± 0.781 | 88.32 ± 0.770 | -0.029 ± 0.016 | 0.0127 ± 0.0022 | 0.0218 ± 0.0029 |
| Pixel-Space | 84.91 ± 0.447 | 83.04 ± 2.900 | 86.77 ± 2.352 | -0.037 ± 0.059 | 0.0168 ± 0.0041 | 0.0239 ± 0.0083 |
| BR-Net (Adeli et al., 2020a) | 81.63 ± 0.499 | 80.26 ± 0.388 | 83.01 ± 0.908 | -0.027 ± 0.011 | 0.0127 ± 0.0006 | 0.0148 ± 0.0002 |
| MDN (Lu et al., 2021) | 87.55 ± 0.6630 | 87.43 ± 3.301 | 87.66 ± 4.277 | -0.002 ± 0.084 | 0.0329 ± 0.0140 | 0.0624 ± 0.0283 |
| P-MDN (Vento et al., 2022) | 86.41 ± 0.876 | 84.25 ± 1.651 | 88.57 ± 1.540 | -0.043 ± 0.030 | **0.0031 ± 0.0009** | 0.0108 ± 0.0017 |
| R-MDN | 85.08 ± 0.591 | 84.98 ± 0.842 | 85.18 ± 1.125 | **-0.002 ± 0.018** | 0.0099 ± 0.0029 | **0.0090 ± 0.0027** |

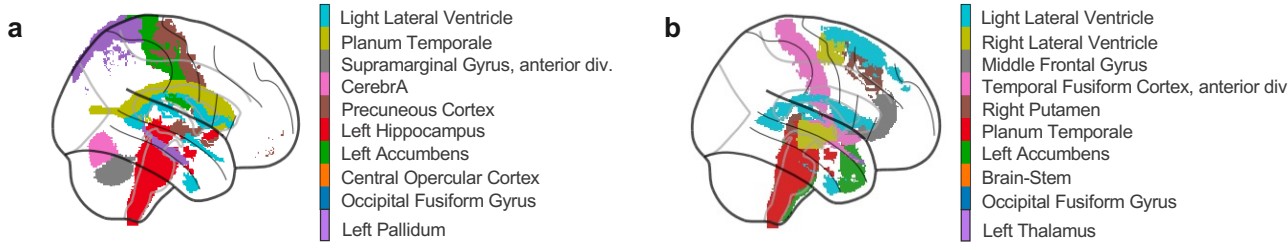

*Figure 6.* **Visualizing ROIs for ABCD sex classification.** The top 10 most relevant regions for distinguishing sex as determined by a model trained **a.** without and **b.** with R-MDN, respectively. One of the ROIs identified by the baseline is the cerebellum, which is implicated in driving sex differences in preadolescence and is the region mostly affected by the confounder for this study. R-MDN does not use the cerebellum for sex classification.

size (Lu et al., 2021), and thus the performance of MDN, but this is a research question orthogonal to our paper.

Further validation of R-MDN in learning confounder-free feature representations is revealed when the model does not use the cerebellum—which is the region mostly confounded by PDS (Adeli et al., 2020b)—for distinguishing sex, while the base model does (Figure 6a,b). These regions are in line with findings in the neuroscience literature about sex differences in preadolescence (Chung et al., 2005; Tiemeier et al., 2010; Fan et al., 2010).

## 5. Conclusion

In this work, we presented Recursive Metadata Normalization (R-MDN)—a flexible layer that can be inserted at any stage within deep neural networks to remove the influence of confounding variables from feature representations. R-MDN leverages the recursive least squares algorithm to operate at the level of individual examples, enabling it to adapt to changing data and confounder distributions in continual learning. It also promotes equitable outcomes across population groups and mitigates catastrophic forgetting of confounder effects over time. As a direction for future work, R-MDN should be explored beyond medical contexts, such as video streams and audio signals, where confounding variables like environmental noise, lighting conditions, camera

angles, or speaker accents might introduce spurious correlations in the data and bias the learning algorithm.

## Acknowledgements

This research was supported in part by the NIH grants R01AG089169, R61AG084471, R01DA057567, and 75N95023C00013. This study was also supported by the Stanford Institute for Human-Centered AI (HAI) Hoffman-Yee award.

## Impact Statement

Computer vision models taken out-of-the-box often generate task predictions that are influenced by the presence of confounding variables in the dataset. These variables, which for instance take the form of patient demographic information, lead to biased predictions. In this paper, we control for confounder influences by proposing Recursive Metadata Normalization (R-MDN), a layer that can be easily integrated in existing models. Upon evaluation, we find that the models trained with R-MDN learn to make equitable predictions across population groups in both continual and static learning settings. We believe that this will encourage the community at large to look towards these models that learn confounder-free feature representations when thinking about deploying systems for society-facing applications.

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

## A. Deriving Parameter Updates for R-MDN

To derive parameter updates for R-MDN, which builds a regression model based on the recursive least squares (RLS) method, we begin from the closed-form solution of ordinary least squares (OLS)-

$$\beta = \left(\sum_{i=1}^{N} X_{i,:} X_{i,:}^{\top}\right)^{-1} \left(\sum_{i=1}^{N} z_i X_{i,:}\right) = R(N)^{-1} Q(N) \tag{8}$$

Firstly,

$$Q(N+1) = Q(N) + z_{N+1} X_{N+1,:} \tag{9}$$

Additionally, using the Sherman-Morrison rank-1 update rule,

$$R(N+1) = \left(R(N) + X_{N+1,:} X_{N+1,:}^{\top}\right)^{-1} = R(N)^{-1} - \frac{R(N)^{-1} X_{N+1,:} X_{N+1,:}^{\top} R(N)^{-1}}{1 + X_{N+1,:}^{\top} R(N)^{-1} X_{N+1,:}} \tag{10}$$

Let

$$P(N+1) = R(N+1)^{-1} = P(N) - K(N+1) X_{N+1,:}^{\top} P(N), \tag{11}$$

where the Kalman Gain

$$K(N+1) = \frac{P(N) X_{N+1,:}}{1 + X_{N+1,:}^{\top} R(N)^{-1} X_{N+1,:}} \tag{12}$$

Rewriting eq. 12,

$$\begin{aligned}
K(N+1) \left[1 + X_{N+1,:}^{\top} P(N) X_{N+1,:}\right] &= P(N) X_{N+1,:} \\
K(N+1) + K(N+1) X_{N+1,:}^{\top} P(N) X_{N+1,:} &= P(N) X_{N+1,:} \\
K(N+1) &= \left[P(N) - K(N+1) X_{N+1,:}^{\top} P(N)\right] X_{N+1,:} \\
K(N+1) &= P(N+1) X_{N+1,:} \text{ [using eq. 11]}
\end{aligned} \tag{13}$$

Finally,

$$\begin{aligned}
\beta(N+1) &= P(N+1) Q(N+1) \\
&= P(N+1) Q(N) + P(N+1) z_{N+1} X_{N+1,:} \text{ [using eq. 9]} \\
&= \left[P(N) - K(N+1) X_{N+1,:}^{\top} P(N)\right] Q(N) + P(N+1) z_{N+1} X_{N+1,:} \text{ [using eq. 11]} \\
&= \left[P(N) - K(N+1) X_{N+1,:}^{\top} P(N)\right] Q(N) + K(N+1) z_{N+1} \text{ [using eq. 12]} \\
&= P(N) Q(N) + K(N+1) \left[z_{N+1} - X_{N+1,:}^{\top} P(N) Q(N)\right] \text{ [using eq. 12]} \\
&= \beta(N) + K(N+1) \left[z_{N+1} - X_{N+1,:}^{\top} \beta(N)\right] \text{ [using eq. 8]} \\
&= \beta(N) + K(N+1) e(N+1),
\end{aligned} \tag{14}$$

where $e(N+1) = z_{N+1} - X_{N+1,:}^{\top} \beta(N)$, the a priori error computed before we update residual model parameters $\beta$.

## B. Computational and Memory Complexity

MDN, P-MDN, and R-MDN each have their tradeoffs in terms of computational complexity, memory complexity, and the extent to which the influence of the confounder is removed from the learned feature representations. As demonstrated in this work, R-MDN empirically works better than both MDN and P-MDN. The asymptotic complexity of each is presented here.

Say there are $N$ training examples, broken into batches of size $B$. Let the confounder matrix $X$ have a shape $N \times p$, where $p$ is associated with the number of confounders, the target, and a bias of 1, and the intermediate learned feature representations have a size of $N \times h$.

Firstly, MDN internally uses the linear least squares estimator, which requires pre-computing the matrix $\Sigma = X^\top X$ in $\mathcal{O}(Np^2)$ steps. Inverting this $p \times p$ matrix further requires $\mathcal{O}(p^3)$ steps. Then, for every batch of information during training, a batch level estimate $\overline{X}^\top \overline{z}$ is produced, where the $\overline{(\cdot)}$ operation refers to a batch instead of the entire training data. This takes $\mathcal{O}(Bph)$ steps. Post-multiplying this $p \times h$ matrix with $\Sigma^{-1}$ requires $\mathcal{O}(p^2 h)$ steps. If computations over batches of information occur $E$ times, the total computational complexity becomes $\mathcal{O}(p^3 + Np^2 + E(p^2 h + Bph))$. In terms of memory complexity, a $p \times p$ $\Sigma^{-1}$ needs to be stored, along with the residual model parameters $\beta$ of size $p \times h$.

For R-MDN, computations only occur over batches of information. In memory, residual model parameters $\beta$ of size $p \times h$ and an estimate of the inverse covariance matrix $P$ of size $p \times p$ are required. For every processing iteration, computing the Kalman gain $K$ requires $\mathcal{O}(Bp^2)$ steps for $PX^\top$, $\mathcal{O}(B^2 p + Bp^2)$ steps for $XPX^\top$, $\mathcal{O}(B^3)$ for inverting this latter matrix, and $\mathcal{O}(B^2 p)$ steps for multiplying the matrices together. Updating $P$ using $KX$ requires $\mathcal{O}(B^2 p)$ steps. And finally, updating $\beta$ requires computing $Ke$ in $\mathcal{O}(Bph)$ steps. The total computational complexity turns out to be $\mathcal{O}(E(B^3 + B^2 p + Bp^2 + Bph))$ steps. Empirically, R-MDN works best with small batch sizes $B$, showing very fast convergence rates, and having a computational complexity that is independent of the size of the training dataset. This becomes important for continual learning, especially longitudinal studies, where data collected over several years or decades can prohibit the use of MDN.

P-MDN does not use a closed-form solution to linear statistical regression. Instead it uses gradient descent to optimize a proxy objective. Thus, the only memory complexity stems from storing $\beta$ parameters of size $p \times h$. The computational complexity is dominated by the number of iterations required to navigate the proxy loss landscape, with results that are not often robust with high variance across runs.

## C. Synthetic Dataset for Static Learning

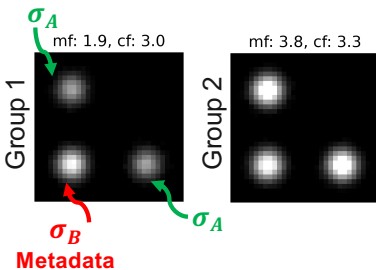

*Figure 7.* A sample from the synthetic dataset used for static learning.

We construct this dataset, following Adeli et al. (2020a) and Lu et al. (2021), by generating 2048 images of size $32{\times}32$, equally divided between two groups (categories). Each image consists of 3 Gaussian kernels: two on the main diagonal, i.e., quadrants 2 and 4, whose magnitudes are controlled by parameter $\sigma_A$, and one on the off-diagonal, i.e., quadrant 3, whose magnitude is controlled by $\sigma_B$ (see Figure 7). Differences in the distributions of $\sigma_A$ between the two groups are associated with the main effects (true discrimination cues) that should be learned by the system, whereas $\sigma_B$ is a confounding variable. An unbiased system will only use information from the main effects for categorization. Both $\sigma_A$ and $\sigma_B$ are sampled from the distribution $\mathcal{U}(1,4)$ for group 1, and $\mathcal{U}(3,6)$ for group 2. Since there is an overlap in the sampling range of the main effects between the two groups, the theoretical maximum accuracy that the system can achieve, were it to not depend on the confounding variable to make discrimination choices, would be $1 - \left(\frac{1}{2}\right)\mathcal{P}[\sigma_A \in \mathcal{U}(3,4)] = 1 - \left(\frac{1}{2}\right)\left(\frac{1}{3}\right) = 0.833$.

We use a 2D convolutional neural network comprising of 2 stacks of convolutions and ReLU non-linearity, followed by 2 fully-connected layers. We apply residualization modules (either MDN, P-MDN, or R-MDN) after every convolution and pre-logits layers (other placement choices explored in Suppl. I). During and after training, we quantify the high-dimensional non-linear correlation between the learned features from the pre-logits layer of the system and the confounding variable through the squared distance correlation ($dcor^2$) metric (Székely et al., 2007). A $dcor^2 = 0$ implies statistical independence between the two distributions.

Our results are summarized in Table 7. We observe that R-MDN consistently reaches the theoretically optimal accuracy and

*Table 7.* **Synthetic dataset results for static learning.** Absolute deviation from the theoretical accuracy $|\text{bAcc} - A|$ ($\downarrow$) and squared distance correlation ($\downarrow$) for various methods and batch sizes. Results are shown over 100 runs of random model initialization seeds with a 95% confidence interval. Best results for each batch size are in bold. There is significant difference in all metrics across all batch sizes for different methods (one-way ANOVA $p < 10^{-58}$). Our method has a significantly better squared distance correlation than MDN for batch sizes less than 128 (post-hoc Tukey's HSD $p < 0.05$) and than P-MDN for batch sizes 2, 64, 256, and 1024 ($p < 0.05$).

| Method | Metric | Batch size | | | | |
|---|---|---|---|---|---|---|
| | | 2 | 16 | 64 | 256 | 1024 |
| Baseline | $|\text{bAcc} - A|$ | $10.49 \pm 0.037$ | $10.49 \pm 0.025$ | $10.50 \pm 0.023$ | $10.52 \pm 0.022$ | $10.55 \pm 0.029$ |
| | $dcor^2$ (group 1) | $0.408 \pm 0.002$ | $0.420 \pm 0.001$ | $0.421 \pm 0.001$ | $0.416 \pm 0.001$ | $0.310 \pm 0.005$ |
| | $dcor^2$ (group 2) | $0.388 \pm 0.003$ | $0.397 \pm 0.001$ | $0.394 \pm 0.001$ | $0.391 \pm 0.001$ | $0.281 \pm 0.005$ |
| MDN | $|\text{bAcc} - A|$ | $8.13 \pm 1.203$ | $4.93 \pm 0.424$ | $3.21 \pm 0.532$ | $\mathbf{0.52 \pm 0.335}$ | $0.95 \pm 0.335$ |
| | $dcor^2$ (group 1) | $0.977 \pm 0.010$ | $0.142 \pm 0.016$ | $0.086 \pm 0.010$ | $0.014 \pm 0.002$ | $\mathbf{0.003 \pm 0.000}$ |
| | $dcor^2$ (group 2) | $0.999 \pm 0.001$ | $0.046 \pm 0.009$ | $0.024 \pm 0.003$ | $\mathbf{0.000 \pm 0.001}$ | $\mathbf{0.000 \pm 0.000}$ |
| P-MDN | $|\text{bAcc} - A|$ | $4.65 \pm 0.448$ | $3.49 \pm 0.373$ | $1.85 \pm 1.151$ | $0.23 \pm 1.361$ | $1.58 \pm 1.983$ |
| | $dcor^2$ (group 1) | $0.042 \pm 0.013$ | $0.022 \pm 0.003$ | $0.050 \pm 0.007$ | $0.048 \pm 0.007$ | $0.098 \pm 0.020$ |
| | $dcor^2$ (group 2) | $0.060 \pm 0.021$ | $0.013 \pm 0.005$ | $0.015 \pm 0.002$ | $0.027 \pm 0.004$ | $0.091 \pm 0.026$ |
| R-MDN | $|\text{bAcc} - A|$ | $\mathbf{0.28 \pm 0.414}$ | $\mathbf{0.04 \pm 0.213}$ | $\mathbf{0.13 \pm 0.088}$ | $1.19 \pm 0.215$ | $\mathbf{0.19 \pm 0.296}$ |
| | $dcor^2$ (group 1) | $\mathbf{0.019 \pm 0.003}$ | $\mathbf{0.014 \pm 0.002}$ | $\mathbf{0.006 \pm 0.001}$ | $\mathbf{0.013 \pm 0.002}$ | $0.015 \pm 0.020$ |
| | $dcor^2$ (group 2) | $\mathbf{0.005 \pm 0.001}$ | $\mathbf{0.001 \pm 0.000}$ | $\mathbf{0.000 \pm 0.000}$ | $0.001 \pm 0.000$ | $0.008 \pm 0.017$ |

a lower $dcor^2$ across all batch sizes. The baseline "cheats" by making use of information from the confounding variable, resulting in a higher balanced accuracy and $dcor^2$. MDN is successful only for large batch sizes like 1024, while being significantly worse for small ones (Lu et al., 2021). While P-MDN is also able to remove the effects of the confounding variable from the learned features to a large extent across all batch sizes (as shown through a smaller $dcor^2$), the large variance across different seed runs suggests that the results are not consistent or robust.

A t-SNE visualization (Van der Maaten & Hinton, 2008) of the learned feature representations shows that the distribution overlap $\mathcal{U}(3, 4)$ for R-MDN is not separable from the two groups for all batch sizes, which means that the system does not use information from the confounding variable for categorization (Figure 8a). In terms of convergence speed, R-MDN does significantly better than both MDN and P-MDN in removing the effects of the confounding variable from the learned features very quickly, especially for small batch sizes (Figure 8b). This effect is attributable to fast convergence properties of the underlying RLS algorithm (Hayes, 1996; Haykin, 2002), and will be advantageous in a continual learning setting when we might not want to train a system until convergence on each training stage, but only for a single or few epochs (read Suppl. L).

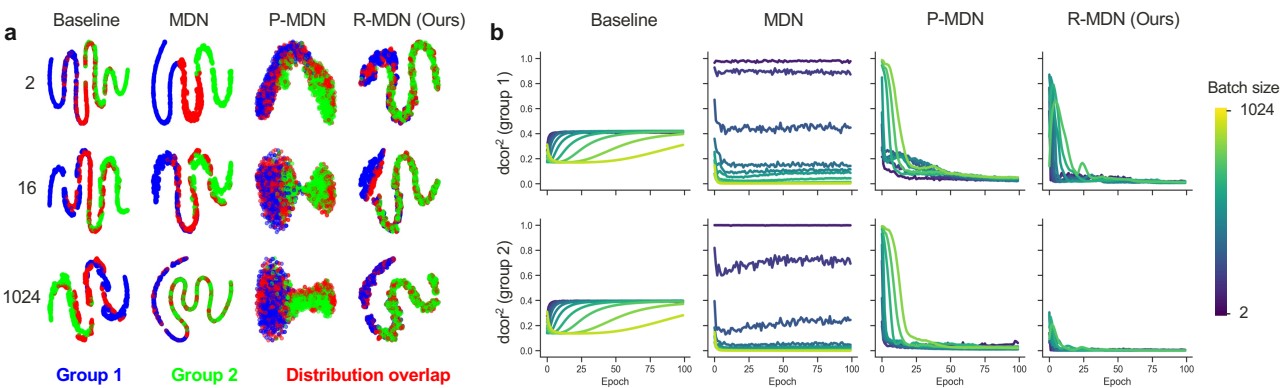

*Figure 8.* **Learned features and squared distance correlation. a.** t-SNE visualization of feature representations for different methods across batch sizes of 2, 16, and 1024. The more separable the distribution overlap $\mathcal{U}(3, 4)$ is in the feature space, the more the method relied on the confounder for discriminating between the groups. **b.** Squared distance correlation across batch sizes for different methods. Each curve represents a different batch size (ranging from 2 to 1024, in increments of powers of 2). Results are shown as the average over 100 runs of random model initialization seeds.

## D. A Second Synthetic Continual Learning Dataset

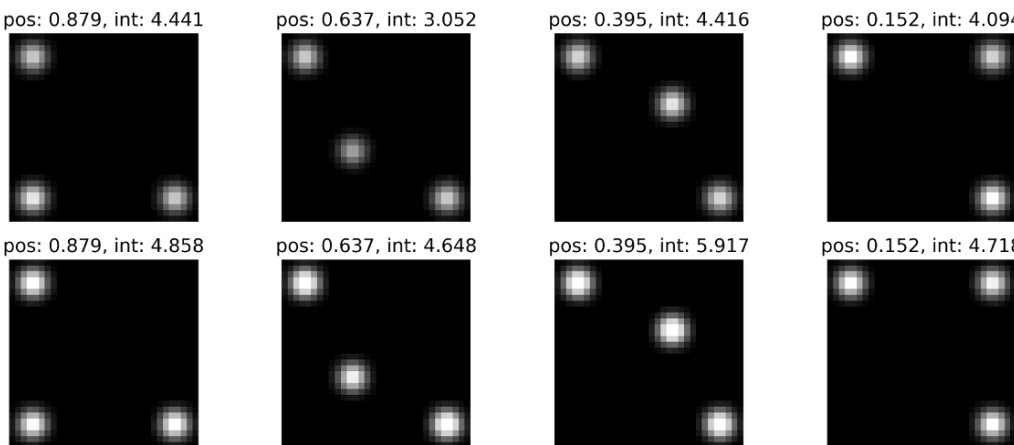

*Figure 9.* **Synthetic continual dataset #2: changing intensities and positions of the confounding variable.** Here we construct a second synthetic dataset for validating our method. We show the change in the distribution of position and intensity of the confounding variable over 4 training stages (moving from left to right on the plot). Similar to Section 4.1, main effects are present on the main diagonal.

While we explore a synthetically generated environment in Section 4.1 to theoretically quantify the influence of confounding variables and the deviation of the model from achieving an un-confounded performance, here we construct a slightly more complex setup wherein we vary two different axes of possibly confounding information: position and intensity. More specifically, we generate 1024 32×32 images that are implicitly broken down into 16 8×8 grids. The top left and bottom right grids contain Gaussian kernels of intensity $\sigma_A$, denoting the main effects. The confounder is represented by a Gaussian kernel of intensity $\sigma_B$, whose position varies from the bottom left to top right of the image over 4 different training stages. Both $\sigma_A$ and $\sigma_B$ are sampled from the distribution $\mathcal{U}(3, 5)$ for group 1, and $\mathcal{U}(4, 6)$ for group 2. A fair classifier should remain unaffected by confounder information, irrespective of its position within the image. Such a setup also allows us to compute theoretical maximum accuracy achievable for each training stage that we can validate methods against. Table 8 presents the results of the experiment, where we can see that R-MDN again allows the model to learn confounder-free features.

*Table 8.* **Results on synthetic continual dataset #2.** The theoretical maximum accuracy for every stage of training is 0.75, which is what an unbiased classifier should achieve. The off-diagonal kernel intensity is taken as the confounder. Results are averaged over 3 runs of random model initialization seeds.

| Method | ACCd | BWTd | FWTd |
|---|---|---|---|
| CNN Baseline | $0.124 \pm 0.002$ | $-0.001 \pm 0.001$ | $0.250 \pm 0.000$ |
| R-MDN | $\mathbf{0.046 \pm 0.031}$ | $\mathbf{0.0006 \pm 0.0009}$ | $\mathbf{0.238 \pm 0.014}$ |

# E. Additional Details on Datasets

## E.1. A Continuum of Synthetic Datasets

This dataset is an extension of the dataset introduced in Suppl. C. For stage 1 across the 3 datasets, we start with the parameter controlling the main effects $\sigma_A \in \mathcal{U}(3,5)$ for group 1, and $\in \mathcal{U}(4,6)$ for group 2. We use the same distributions for $\sigma_B$ (which controls the magnitude of the confounding variable). This means that prediction kernels (i.e., those associated with true discrimination cues) are more "intense" (have a higher magnitude) for images in group 2 than in group 1.

**Dataset 1: Confounding variable distribution changes.** We keep the distribution of main effects constant but vary that of the confounding variable across different stages. With every new stage, we decrease the entire range of $\sigma_B$ by 0.125 for group 1, and increase it by the same amount for group 2. For an unbiased classifier that uses short-cut learning by focusing on the confounder distribution, the problem becomes easier with "time" and performance will likely increase. This is what we observe with the baseline model, which has the same architecture as that in Section C (see Figure 12c).

**Dataset 2: Distribution of main effects changes.** Next, we keep the distribution of the confounding variable constant and vary that of the main effects across different stages instead. In contrast to the above, with every new stage, we increase the entire range of $\sigma_A$ by 0.125 for group 1, and decrease it by the same amount for group 2. This results in the problem becoming more difficult with "time". Performance for an unbiased classifier should drop for later tasks during learning.

**Dataset 3: Distributions of both the confounding variable and main effects change.** This dataset is a combination of the above two, with the difference in the distribution of the confounding variable across the two groups becoming more pronounced with "time", while that of the main effects starting to become more similar.

## E.2. HAM10000 Dataset

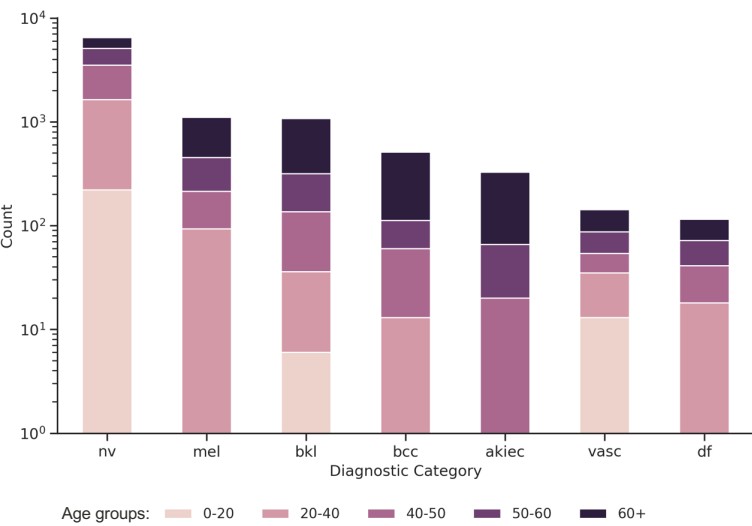

*Figure 10.* Distribution of dermatoscopic images across different diagnostic categories for various age brackets in the HAM10000 dataset.

The *Human Against Machine with 10000 training images* (HAM10000) dataset (Tschandl et al., 2018) is a multi-source collection of 10015 dermatoscopic images for diagnosis of common pigmented skin lesions. These images have been collected from different populations through different modalities. Diagnostic categories include:

- **akiec:** actinic keratoses and intraepithelial carcinoma or Bowen's disease

- **bcc:** basal cell carcinoma

- **bkl:** benign keratosis-like lesions (solar lentigines or seborrheic keratoses and lichen-planus like keratoses)

- **df:** dermatofibroma

- **mel:** melanoma

- **nv:** melanocytic nevi

- **vasc:** vascular lesions (angiomas, angiokeratomas, pyogenic granulomas and hemorrhage)

Lesions were confirmed either through histopathology, follow-up examinations, expert consensus, or in-vivo confocal microscopy.

Figure 10 shows the distribution of age for various diagnostic categories. The change in the age distribution is significant. Age is a confounder for this dataset because it affects both the target categories (certain categories like melanoma mostly occur in older patients) and the input images (skin appearance might change with age).

### E.3. ADNI Dataset

The Alzheimer's Disease Neuroimaging Initiative (ADNI) (`adni.loni.usc.edu`) was launched in 2003 with the goal to test whether serial MRI, PET, other biological markers, and clinical and neurophysiological assessments can be used to measure the progression of mild cognitive impairment (MCI) and early Alzheimer's disease (AD) (Mueller et al., 2005; Petersen et al., 2010). There are a total of 3880 T1w MRIs from 573 participants across years 2006 through 2017.

The distribution of age and sex across subjects diagnosed with MCI or AD, and the control group (CN) is shown in Figure 11 and Table 9.

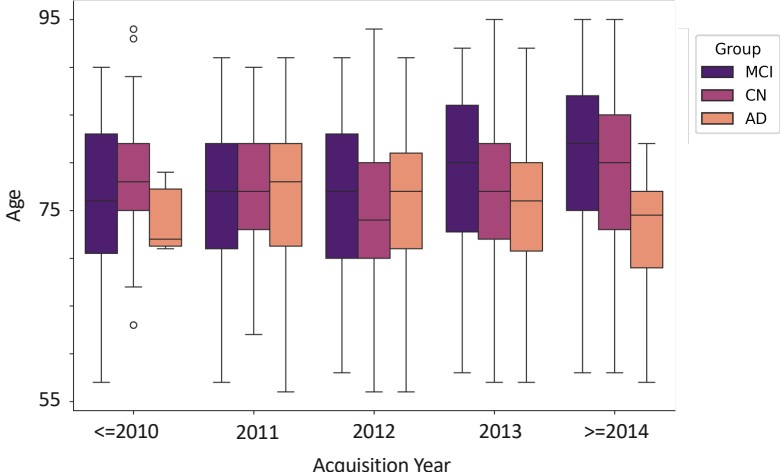

*Figure 11.* Distribution of age across different participant groups in the ADNI dataset.

*Table 9.* **Sex distribution across different groups in the ADNI study**. Differences are statistically significant, as measured via a $\chi^2$-test.

| Group | Sex | Year | | | | |
|---|---|---|---|---|---|---|
| | | **2010** | **2011** | **2012** | **2013** | **2014** |
| AD | Male | 2 | 52 | 214 | 125 | 20 |
| | Female | 4 | 26 | 110 | 95 | 16 |
| CN | Male | 71 | 238 | 410 | 208 | 206 |
| | Female | 81 | 222 | 402 | 221 | 197 |
| MCI | Male | 117 | 145 | 140 | 81 | 98 |
| | Female | 82 | 84 | 106 | 55 | 52 |

### E.4. ABCD Study

The Adolescent Brain Cognitive Development (ABCD) study (`https://abcdstudy.org`) is a multisite, longitudinal study. More than 10,000 boys and girls from the U.S. between the ages of 9-10 were recruited that were diverse in

terms of race/ethnicity, education and income levels, and living environments (Thompson et al., 2019). Garavan et al. (2018) provide a more detailed account of the population neuroscience approach to recruitment and inclusion/exclusion criteria. Appropriate consent was requested before participation in the ABCD study. Data is anonymized and curated, and is released annually to the research community through the NIMH Data Archive (see data sharing information at https://abcdstudy.org/scientists/data-sharing/). The ABCD data repository grows and changes over time. The ABCD data used in this report came from release 5.0, with DOI 10.15154/8873-zj65.

Table 10 shows the distribution of participants (boys and girls) in the study with respect to age, pubertal development score (PDS), and race. PDS is significantly larger for girls than boys, and thus serves as a confounder for this study.

*Table 10.* **Variable distributions across boys and girls in the ABCD study**. Mean and standard deviation for age and pubertal development scale (PDS), and the number of subjects of each race in the study across boys and girls. PDS is an integer between 1-5. Differences are significant across boys and girls for PDS (measured using a two-sample t-test) but not age or race. Girls have a higher PDS than boys in the study. All values are for the first visit of each subject.

|  |  | **Boys** | **Girls** | **p-value** |
|---|---|---|---|---|
| Age (in months) |  | $119.18 \pm 7.562$ | $118.80 \pm 7.525$ | $>0.001$ |
| PDS |  | $1.372 \pm 0.622$ | $2.182 \pm 0.907$ | $<0.001$ |
| Race/ethnicity | White | 2953 | 2659 |  |
|  | Black | 702 | 735 |  |
|  | Hispanic | 1060 | 999 | $>0.001$ |
|  | Asian | 103 | 114 |  |
|  | Other | 567 | 529 |  |

# F. Additional Details on Methods

All experiments were run on a single NVIDIA GeForce RTX 2080 Ti with 11GB memory size and 8 workers on an internal cluster.

## F.1. Synthetic Dataset for Static Learning

The base model was a CNN consisting of two convolutional layers followed by two fully connected layers. The first convolutional layer had 16 output channels and a kernel size of 5, the second had 32 output channels and a kernel size of 5, and the pre-logits fully connected layer had a hidden size of 84. Models were trained for 100 epochs with different batch sizes. Parameters of the R-MDN model were optimized using Adam (Kingma & Ba, 2014), with a learning rate initialization of 0.0001 that decayed by 0.8 times every 20 epochs. The regularization parameter for R-MDN was set to 0.0001.

## F.2. ABCD Sex Classification

Raw MRI images were downloaded, skull-stripped, and affinely registered to the MNI 152 template (Mazziotta et al., 1995; 2001a;b). Data augmentation involved removing MRIs for all subjects that did not have an associated PDS score recorded. We downscaled all MRIs to $64 \times 64 \times 64$ volumes, performed random one voxel shift and one degree rotation in all three Cartesian directions, and random left-right flip (since sex affects the brain bilaterally (Hill et al., 2014; Hirnstein et al., 2019)) for training images. To evaluate the models, we perform 5 runs of 5-fold cross validation across different model initialization seeds, with images split by subject and site ID, and having approximately an equal number of boys and girls in each fold.

The base model was a CNN consisting of three convolutional layers, each followed by max pooling, and two fully connected layers. The first convolutional layer had 8 output channels with a kernel size of 3, the second had 16 output channels with a kernel size of 3, and the third had 32 output channels with a kernel size of 3. The pre-logits fully connected layer had a hidden size of 32. For max pooling, the first and second layers used a kernel size of 2 with a stride of 2, while the third layer had a kernel size of 4 with a stride of 4. Models were trained for 50 epochs with a batch size of 128. Parameters of the R-MDN model were optimized using Adam, with a learning rate initialization of 0.0005 that decayed by 0.7 times every 4 epochs. The regularization parameter for R-MDN was set to 0.

## F.3. A Continuum of Synthetic Datasets

Model architecture used is the same as that for the static learning setting. Models were trained for 100 epochs with a batch size of 128. Parameters of the R-MDN model were optimized using Adam, with a learning rate initialization of 0.0005 that decayed by 0.8 times every 20 epochs. The regularization parameter for R-MDN was set to 0.0001.

## F.4. HAM10000 Skin Lesion Classification

The dataset was first downsampled to $64 \times 64 \times 64$ and then divided into five training stages based on age groups: $< 20$, $[20, 40)$, $[40, 50)$, $[50, 60)$, and $\geq 60$.

- **Stage 1**: 50% of the images came from $< 20$, 30% from $[20, 40)$, 10% from $[40, 50)$, 5% from $[50, 60)$, and 5% from $\geq 60$.

- **Stage 2**: 5% from $< 20$, 50% from $[20, 40)$, 30% from $[40, 50)$, 10% from $[50, 60)$, and 5% from $\geq 60$.

- **Stage 3**: 5% from $< 20$, 5% from $[20, 40)$, 50% from $[40, 50)$, 30% from $[50, 60)$, and 10% from $\geq 60$.

- **Stage 4**: 10% from $< 20$, 5% from $[20, 40)$, 5% from $[40, 50)$, 50% from $[50, 60)$, and 30% from $\geq 60$.

- **Stage 5**: 30% from $< 20$, 10% from $[20, 40)$, 5% from $[40, 50)$, 5% from $[50, 60)$, and 50% from $\geq 60$.

The base model was a ViT with a patch size of 8, 12 hidden layers, 12 heads, a hidden dimension of 384, an MLP dimension of 1536, and the hidden size for the pre-logits layer as 96. Models were trained for 30 epochs with a batch size of 128. Parameters of the R-MDN model were optimized using AdamW (Loshchilov & Hutter, 2017), with a learning rate initialization of 0.0005 that decayed by 0.8 times every 5 epochs. We imposed a weight decay of 0.001. The regularization parameter for R-MDN was set to 0.00001.

### F.5. ADNI Diagnostic Classification

The dataset was first downsampled to $64\times64\times64$ and then divided into five training stages based on acquisition year bins: $\leq$2010, 2011, 2012, 2013, and $\geq$2014.

The base model was a 3d ViT with a patch size of 16, 6 hidden layers, 8 heads, a hidden dimension of 384, an MLP dimension of 1536, and the hidden size for the pre-logits layer as 768. Models were trained for 50 epochs with a batch size of 128. Parameters of the R-MDN model were optimized using AdamW, with a learning rate initialization of 1e-05 that decayed according to a cosine annealing scheduler. We imposed a weight decay of 1e-06. The regularization parameter for R-MDN was set to 0.

# G. Visualizing Results for Continual and Static Learning

Here we visualize the results we present through tables in Sections 4.1, 4.2, and 4.4.

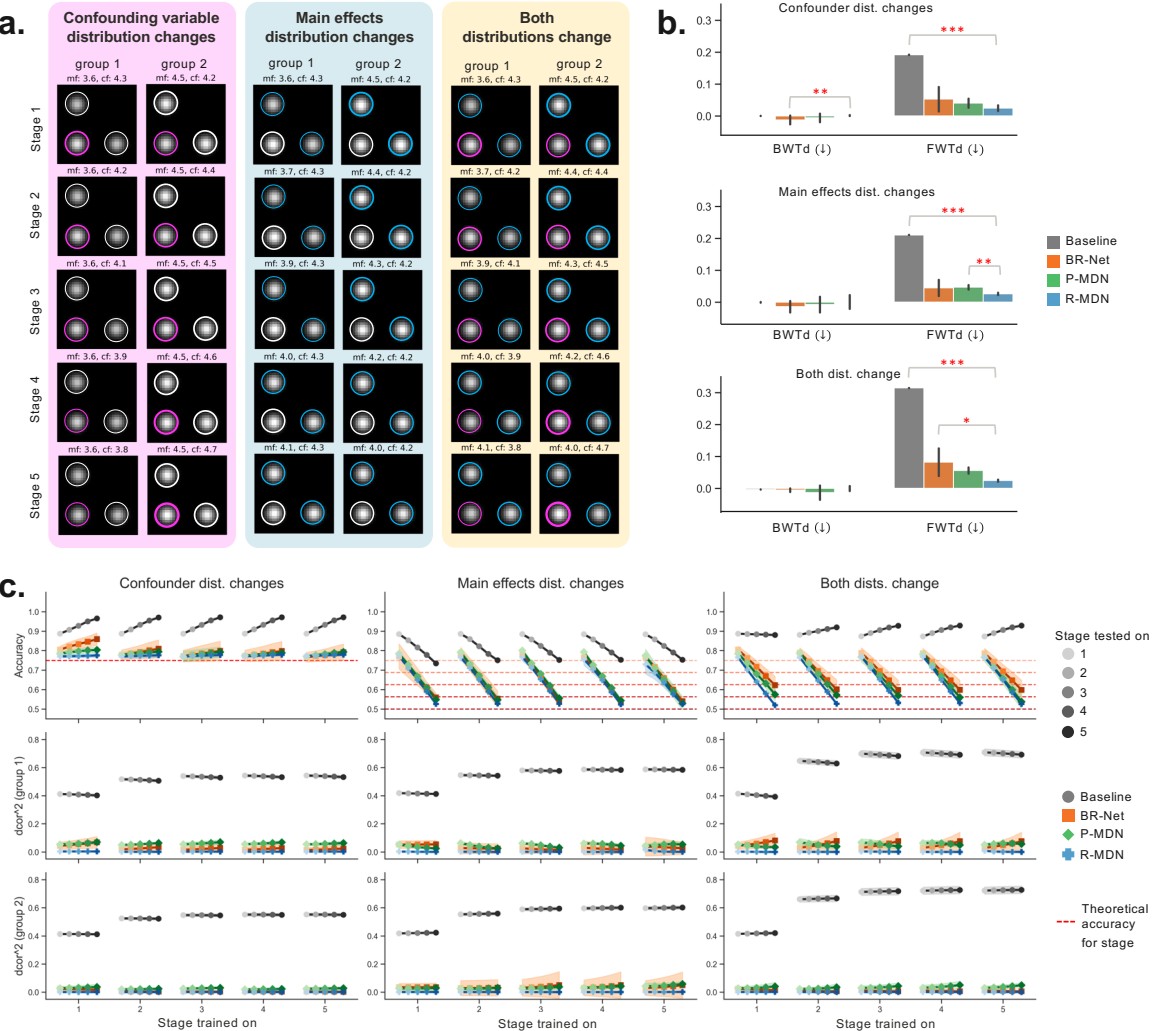

*Figure 12.* **Synthetic dataset results for continual learning. a.** Samples from the synthetic datasets used for continual learning. We annotate main effects and confounders with boundaries of different widths to visually aid in distinguishing between their magnitudes. **b.** BWTd and FWTd mean and standard deviation for different methods and datasets. The closer the bar to 0, the better the model. A total of 5 runs were performed with different model initialization seeds. A post-hoc Conover's test with Bonferroni adjustment was performed between those groups of methods where a Kruskal-Wallis test showed significant differences ($p < 0.05$). **c.** Accuracy and squared distance correlation for different methods and datasets. For each stage that the model is trained on, it is evaluated against the test sets of all 5 stages (shown through solid curves). Less opaque markers represent earlier stages, while more opaque markers represent later stages being evaluated on. Dotted red lines of various transparency values show the theoretical maximum accuracy that an unbiased model will get for each of the different stages.

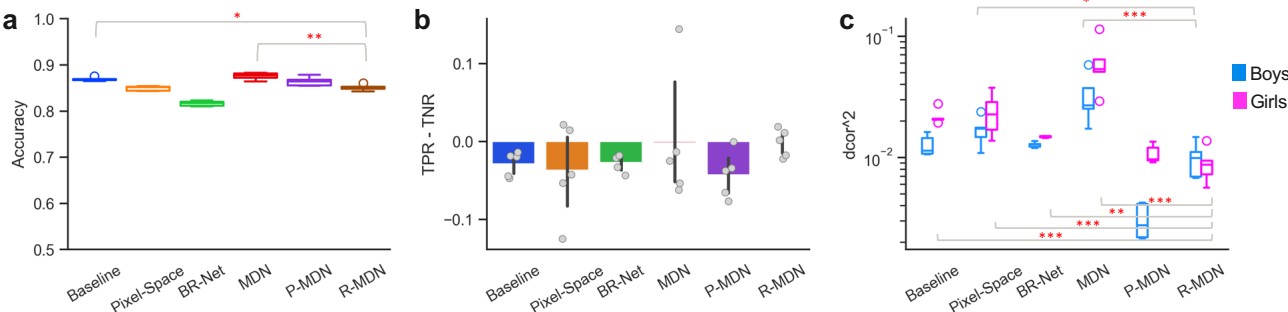

*Figure 13.* **Visualizing different metrics for the ABCD dataset. a.** Accuracy, **b.** difference between True Positive Rate (TPR) and True Negative Rate (TNR), and **c.** dcor$^2$ between learned features and PDS for boys and girls for different methods. Results shown over 5 folds of 5-fold cross validation, with data split by subject and site ID. Statistically significant differences between R-MDN and other methods are measured first using Kruskal-Wallis and then a post-hoc Conover's test with Bonferroni adjustment.

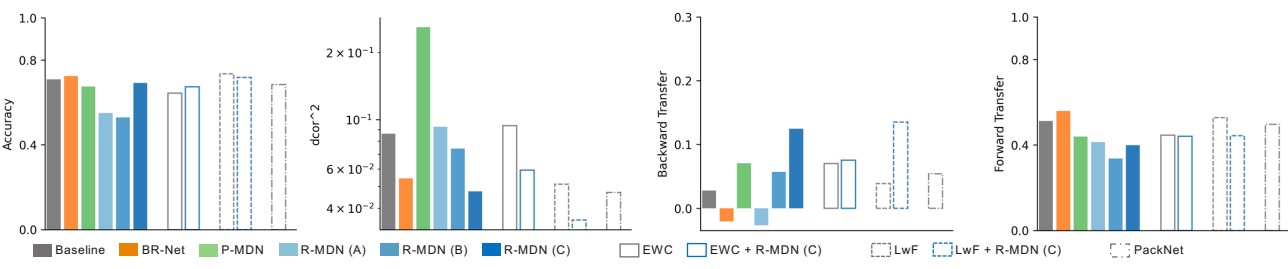

*Figure 14.* **Visualizing different metrics for HAM10K skin lesion classification.** Accuracy, squared distance correlation, backward transfer, and forward transfer for different methods. Results are shown after training each model on the final training stage.

# H. Effect of Regularization Hyperparameter

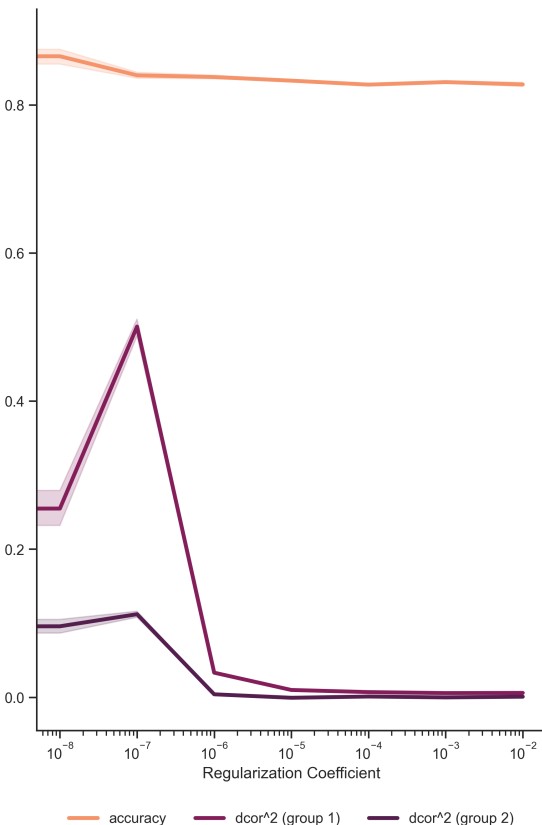

*Figure 15.* Accuracy and squared distance correlation when the regularization hyperparameter for the R-MDN module is varied. Results are computed for the synthetic dataset described in Section C, and we show the mean and 95% CI over 100 runs of random model initialization seeds.

In Figure 15, we systematically vary the regularization hyperparameter $\lambda$ to assess how sensitive model performance and the ability to learn confounder-independent feature representations are to its value. We observe that model performance remains consistently robust across different values of $\lambda$. However, we find that the capacity to residualize the confounder's effects improves with higher values of $\lambda$, probably due of it stabilizing the residualization process.

## I. R-MDN Module Placement Choice

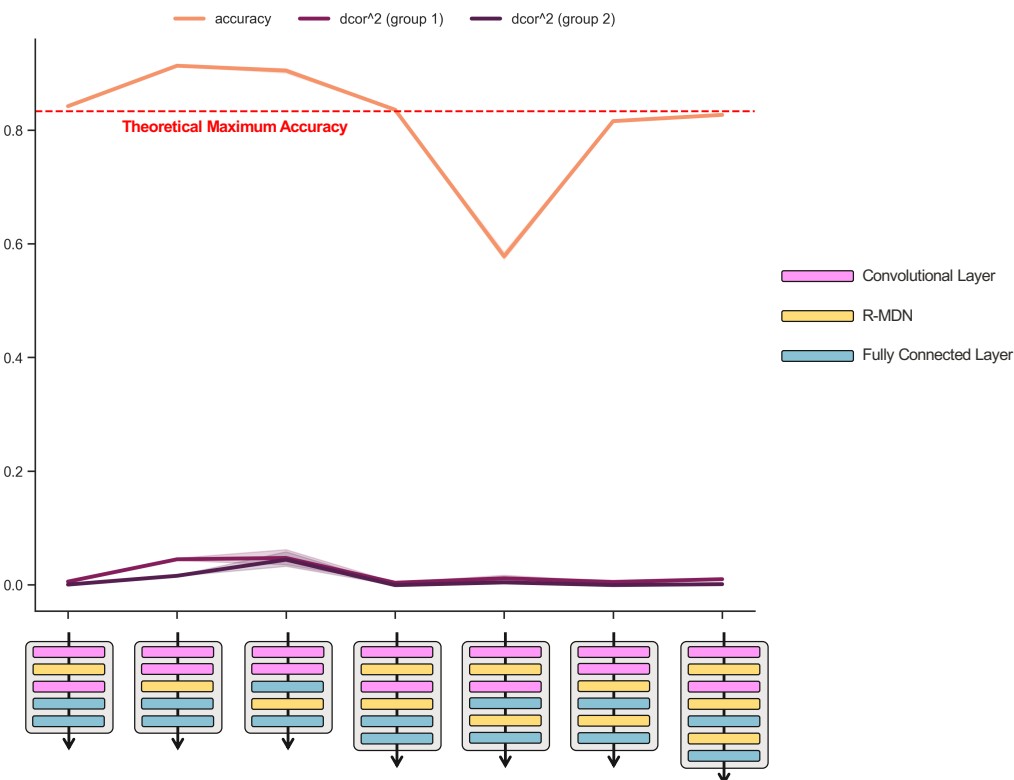

*Figure 16.* Accuracy and squared distance correlation when the R-MDN module is inserted at various locations in a convolutional neural network. Results are computed for the synthetic dataset described in Section C, and we show the mean and 95% CI over 100 runs of random model initialization seeds.

In Figure I, we vary the placement of the R-MDN layers within a deep convolutional neural network to observe the effects on model performance and correlation of the learned features with the confounder. We find that while model performance seems to be sensitive to the placement, the ability to remove the influence of the confounder from the feature representations is, overall, consistently high. For such an architecture, adding an R-MDN layer after every convolutional layer and the pre-logits layer seems to provide the best trade-off between model performance and residualization (as also observed by Lu et al. (2021)).

## J. Glass Brain Visualizations for ABCD Sex Classification

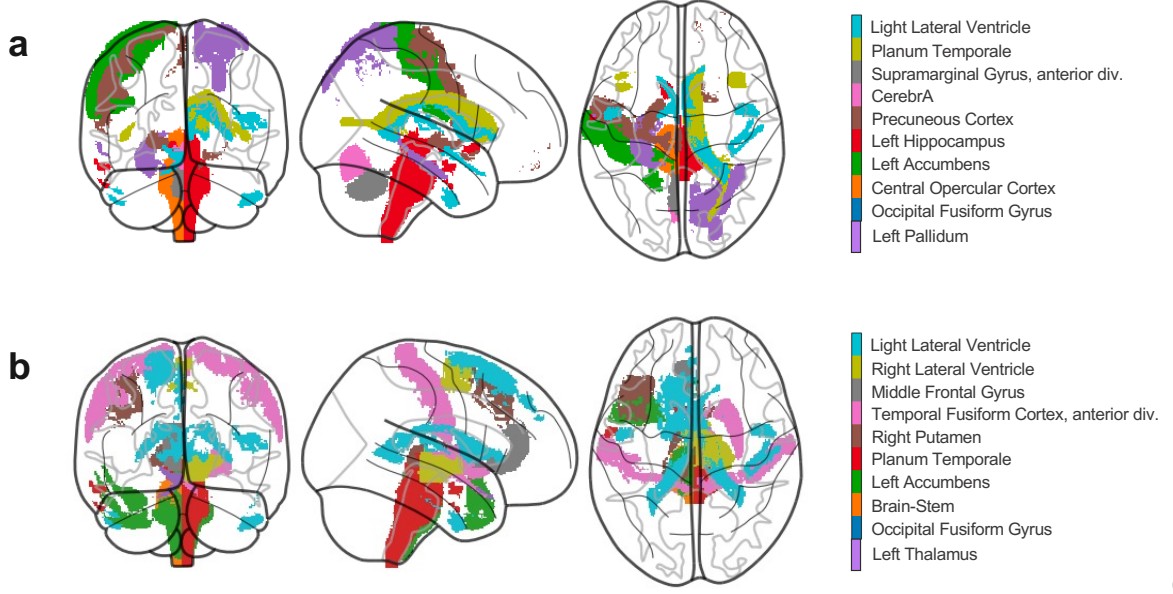

*Figure 17.* The top 10 regions identified as being relevant for distinguishing sex by **a.** the base model, and **b.** the same model trained with R-MDN.

To identify the top 10 most relevant regions for distinguishing sex (Figure J), we first generate 3D saliency maps based on the test set images, highlighting areas in the input image that most activate the model. A threshold of 0.05 is applied to focus on the most salient regions. A 5x5x5 smoothing filter is applied, replacing each voxel's value with the average of its neighboring voxels. These regions are then visualized using the Harvard atlas.

# K. Quantifying Sensitivity To Confounders

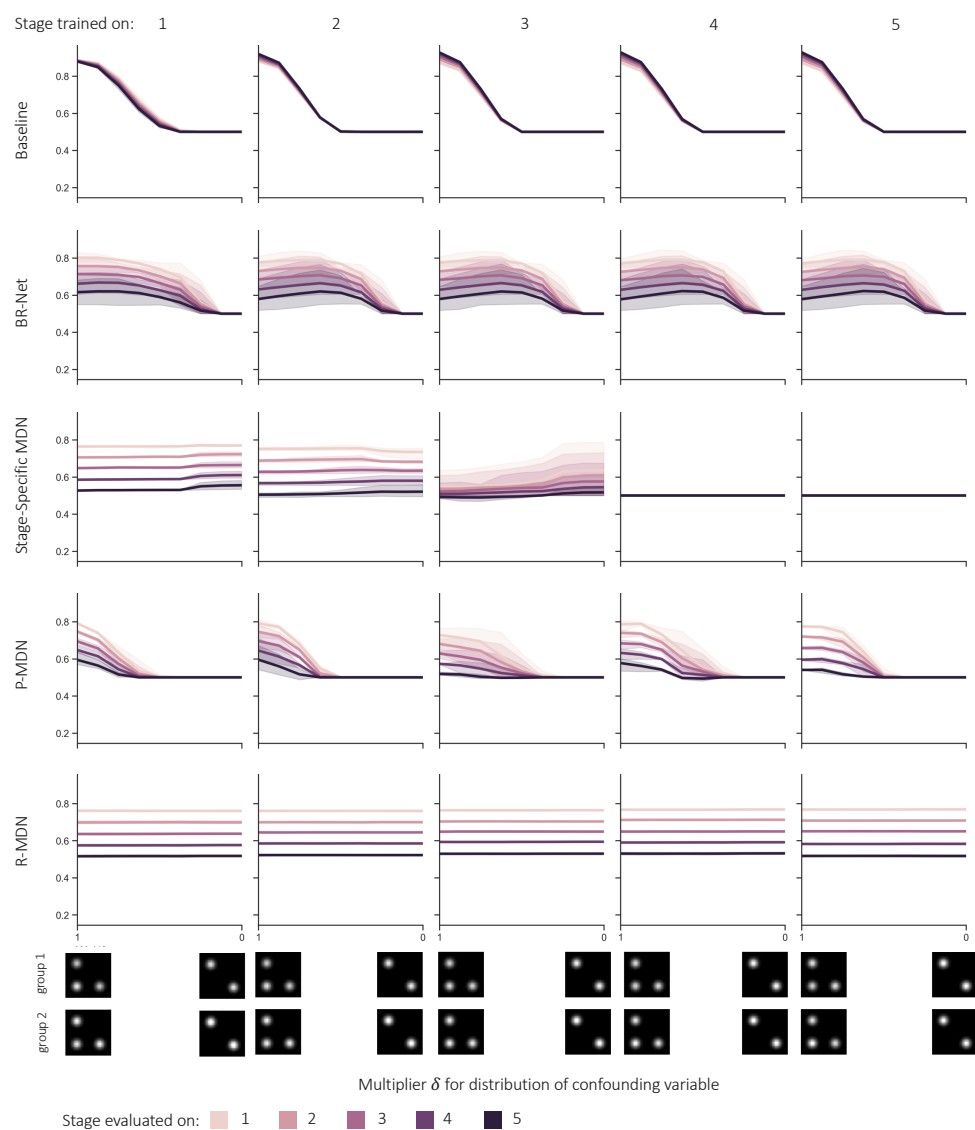

*Figure 18.* Accuracy for various methods get in a continual learning setting when evaluated on test sets with various distributions of the confounding variable. Each row represents a different method, each column the stage that the model is trained on after which it is evaluated, and each hue of the curve the stage that the model is evaluated on. A $\delta = 0$ implies that that input does not contain a confounder. Results are evaluated on the synthetic dataset from Section 4.1 where we change the distributions for both the confounding variable and main effets. We show the mean and 95% CI over 3 runs of random model initialization seeds.

In Figure 18, we provide additional plots for the experiment visualized in Figure 3d. We vary the intensity of the confounder by applying a multiplier $\delta \in [0, 1]$.

## L. Effect of Training Protocol for Continual Learning

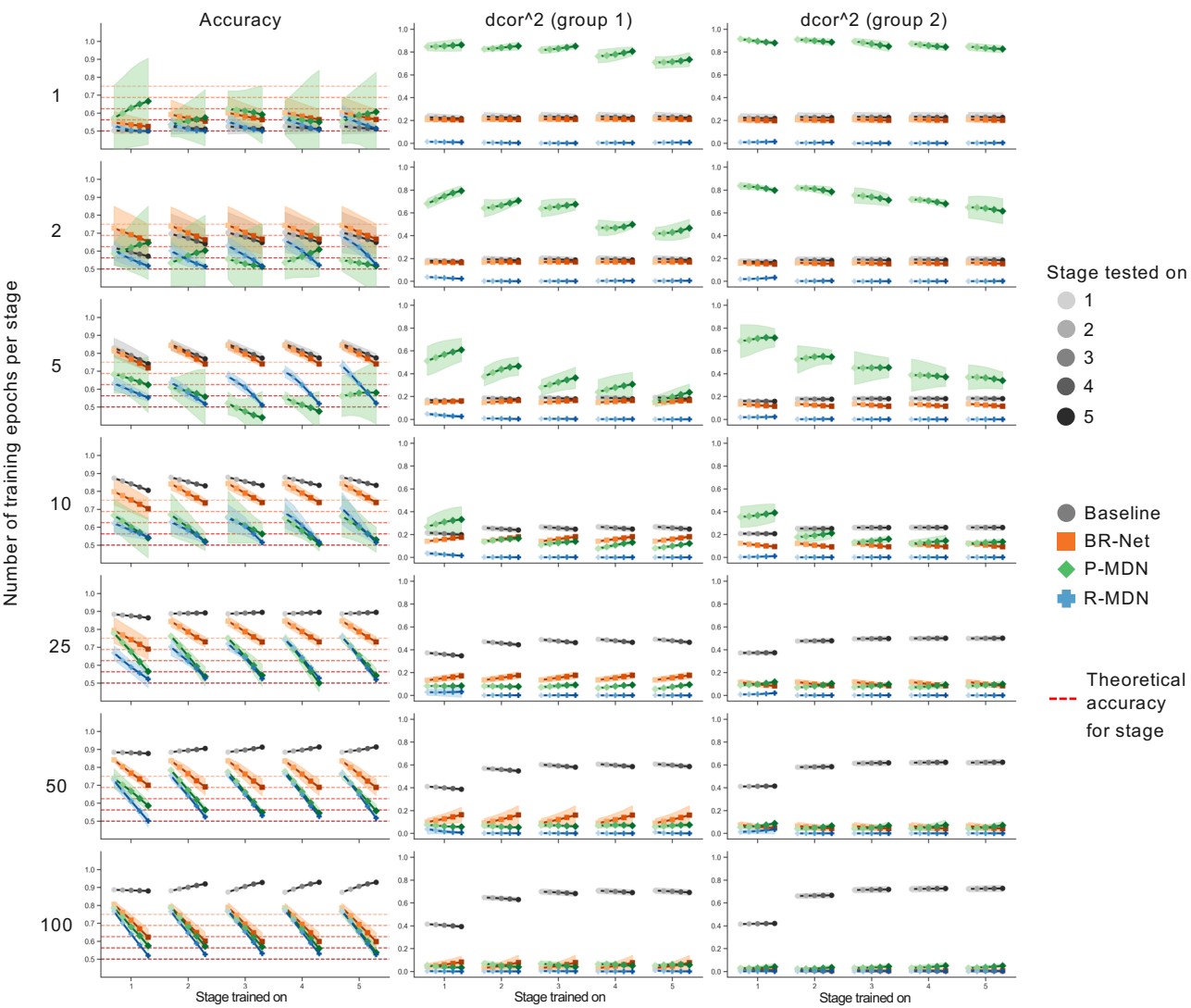

*Figure 19.* Accuracy and squared distance correlation for different methods and number of training epochs per stage. We used the synthetic dataset where the distributions for both the confounding variable and main effects change. For each stage that the model is trained on, it is evaluated against the test sets of all 5 stages (shown through solid curves). Less opaque markers represent earlier stages, while more opaque markers represent later stages being evaluated on. Dotted red lines show the theoretical maximum accuracy that an unbiased model will get for each of the different stages. Results shown as the mean and 95% CI over 5 runs.

Here, we quantify how task performance and the ability to learn confounder-free feature representations change with different number of training epochs per stage; i.e., with the number of times every example from the training data is presented to the system. We observe that R-MDN is the only method that is able to remove the influence of the confounder from the learned features for smaller number of training epochs. This is perhaps because of R-MDN's fast convergence abilities (Hayes, 1996; Haykin, 2002)—a property that gradient- and adversarial-based methods are not able to demonstrate (Figure 19). This is further reinforced by R-MDN having a better forward transfer on future stages of training for both small and large numbers of training epochs (Figure 20). Both BR-Net and P-MDN are decent methods for continual learning, but they require the same training examples to be seen multiple times in order to drive high prediction scores and remove confounder influence from learned features.

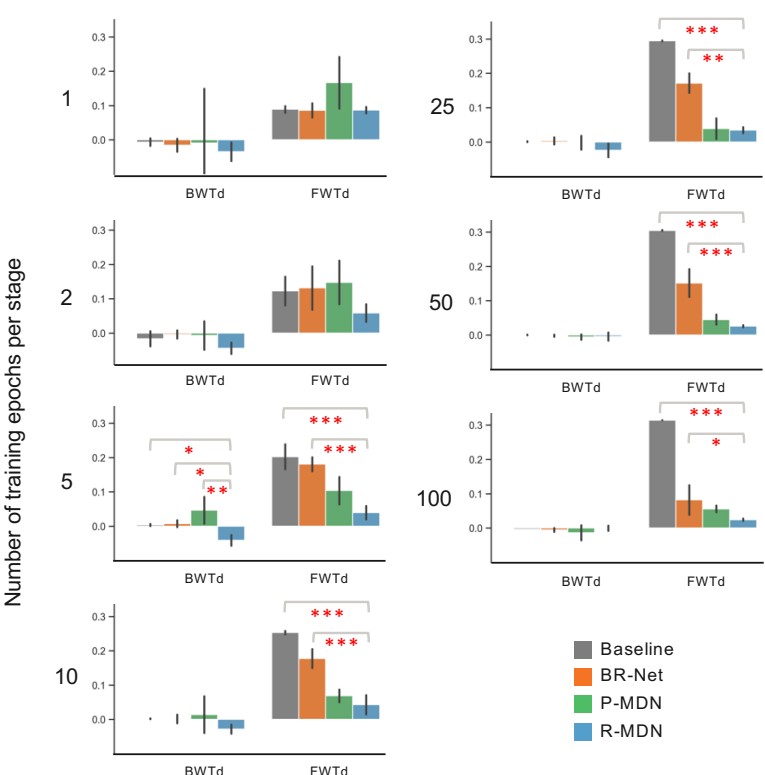

*Figure 20.* BWT and FWT mean and standard deviation for different methods and number of training epochs per task. We used the synthetic dataset where the distributions of both the confounding variable and main effects change. The closer the bar to 0, the better the model. A total of 5 runs were performed with different model initialization seeds. A post-hoc Conover's test with Bonferroni adjustment was performed between those groups of methods where a Kruskal-Wallis test showed significant differences ($p < 0.05$).

