# OpenReview forum: "Confounder-Free Continual Learning via Recursive Feature Normalization"
_ICML.cc/2025/Conference — ICML 2025 poster_

### Official Review · Reviewer_97mW · 2025-03-09

**Overall Recommendation:** 3

**Summary:**

This paper introduces a Recursive metadata normalization (R-MDN) layer, in order to remove confounders, that are extraneous variables that affectboth the input and the target. The paper extends the confounder-removing activity to continual learning community. R-MDN performs statistical regression via the recursive least squares algorithm to maintain and continually update an internal model state
with respect to changing distributions of data and confounding variables.

**Claims And Evidence:**

The claims in this paper are well supported in both methology and experiment aspects.

**Essential References Not Discussed:**

This paper discusses the recursive learning method in continual learning. There is a new branch of continual learning, named analytic continual learning (starting from ACIL [1]), which also adopts recursive merhod, such as recursive least squares related techniques. The papers in this branch shares several similarities in methology (from derivation to result).

[1] ACIL: analytic class-incremental learning with absolute memorization and privacy protection

**Experimental Designs Or Analyses:**

The experiment settings are overall quite good

**Methods And Evaluation Criteria:**

In the main context, the paper only reports results in medical datasets. It seems that the method should cover various datasets (at least from the title and abstract). The reviewer concerns that the proposed method cannot handle large-scale problems, e.g., ImageNet-1000.

**Other Comments Or Suggestions:**

N.A

**Other Strengths And Weaknesses:**

See the comments marked above.

**Questions For Authors:**

The use of RLS in updating the normalization layer seems interesting. Please state the difference between you method and the analytic continual learning techniques. Could it be a trivial implementation? If not, please do make a thorough comparison. Please note that my rating is not a final decision.

##after rebuttal## according to the response, I am maintaining the rating.

**Relation To Broader Scientific Literature:**

N/A

**Theoretical Claims:**

The proofs for theoretical claims seems valid.

---

> ### Author Rebuttal · Authors · 2025-04-01
>
> Thank you for your thorough review and thoughtful feedback—we respond below to the concerns you have raised.
>
> ---
>
> > The paper only reports results in medical datasets. It seems that the method should cover various datasets (at least from the title and abstract). The reviewer concerns that the proposed method cannot handle large-scale problems, e.g., ImageNet-1000.
>
> Limitations of the methods on dataset size are definitely an important consideration. In this paper, we work within the framework of medical datasets for real-world setups (wherein issues with confounded learning is very prominent), and non-medical datasets for synthetic setups. We build on prior works from Zhao et al. (2020), Adeli et al. (2020a), Lu et al. (2021), and Vento et al. (2022), who have proposed methods within this framework that we can directly compare to. Several of these datasets are medium-sized, wherein our method successfully learns confounder-free representations, as demonstrated empirically. Larger problems manifest through either larger dataset sizes, longer pre-training, or both. We evaluate the effect of the number of training epochs on confounder-free learning in Suppl. K, where R-MDN’s quick convergence property becomes an advantage. For larger dataset sizes, catastrophic forgetting becomes an issue with task learning. We make preliminary advances in this direction in Exp. 4.2 by integrating R-MDN with existing continual learning frameworks such as LwF and EWC that deal with catastrophic forgetting. Memory replay is another technique that our method could be integrated with. Nevertheless, we acknowledge this issue, and implementing and analyzing plausible solutions is an exciting direction for future work.
>
> ---
>
> > Please state the difference between your method and the analytic continual learning techniques. Could it be a trivial implementation? If not, please do make a thorough comparison.
>
> Thank you for referencing the ACIL paper. ACIL works within the domain of class-incremental continual learning by breaking down training into two stages: (a) base training via backpropagation, which allows learning of model parameters for the task; and (b) analytic re-alignment, which maps intermediate model features onto the label matrix recursively. This recursive construction permits absolute memorization and effectively mitigates catastrophic forgetting. Our approach is different since we map intermediate features not onto the label space but onto the confounder space so that the residual component of the features can be extracted and passed to downstream layers. This effectively removes the influence of such confounding variables from learned feature representations. Additionally, everything happens in a single-stage end-to-end framework for R-MDN: no network parameters are frozen for the recursive updates. Thus, the network has to learn task-relevant features that are also confounder-free during base training. Overall, the implementation nuance (single-stage learning), the target for the recursive least squares estimator (confounding variables), as well as the input to downstream layers (the residual) differ between our method and ACIL. We will include these details in the final paper.

---

> > ### Comment · Reviewer_97mW · 2025-04-03
> >
> > Thank you for the response. There a still a few concerns:
> >
> > 1) If you deal with medical data only, the title and abstract are too general. It is normal that it is treated as a regular continual learning paper and asked to compared in SOTA fashion.
> >
> > 2) This method resemble analytic continual learning series closely. If not for ACIL, you may try F-OAL in dealing with online tasks which are more close to your task.
> >
> > F-OAL: Forward-only Online Analytic Learning with Fast Training and Low Memory Footprint in Class Incremental Learning
> >
> > I am lowing the raiting at the moment and it would be subject to increase again.

---

> > > ### Author Response · Authors · 2025-04-04
> > >
> > > Thank you for the comment.
> > >
> > > ---
> > >
> > > > This method resemble analytic continual learning series closely. If not for ACIL, you may try F-OAL in dealing with online tasks which are more close to your task.
> > >
> > > While we understand the reviewer's concern, as F-OAL has some similarities to our method in terms of using recursive least squares, it addresses an inherently different problem. With R-MDN, we are working within the **framework** of continual learning to learn feature representations that are not just task-specific (as extensive research in the past with works such as LwF, EWC, PackNet, ACIL, F-OAL, etc. have done) but also invariant to bias in the form of confounding variables present in the dataset. Thus, the aim of the work is not to come up with a new technique to avoid catastrophic forgetting, which past works mentioned before have extensively explored, but to build **on top of** them to **additionally** remove the influence of confounders from the features.
> > >
> > > R-MDN is a normalization layer and not a learning algorithm for model training. It computes the residual of the features that are confounder-invariant and passes it to the downstream layer. On the other hand, methods such as ACIL and F-OAL are proposed to **train** models in the setting of class-incremental learning. Their aim is to address catastrophic forgetting and improve **task accuracy**. We care about **how biased** the learned features are during training and how might we overcome that. High accuracy does not mean unbiased learning, as we have shown throughout the paper. Since R-MDN and works such as LwF, EWC, ACIL, F-OAL, etc. are **orthogonal** research directions. This means that R-MDN also be used **on top of** those methods to ensure minimzation of the effects from bias and confounders. Subsequently, we have shown through Exp. 4.2 and Table 4.2 that we can add R-MDN to LwF and EWC to promote confounder-free learning. Similarly, R-MDN can also be complemented with ACIL and F-OAL, since, reiterating, R-MDN is a normalization layer that is added to the model architecture, whereas F-OAL is a training framework for models. While both F-OAL and R-MDN might be building some variant of a recursive least squares regressor, F-OAL **does not** learn confounder-free representations, as shown through this table. To address the reviewer's concerns, we conducted additional experiments on the non-medical synthetic continual dataset we presented in the paper (Section 4.1), where both the confounders and main effects change (over 3 runs of random model initialization seeds):
> > >
> > > | Method | ACCd (&#8595;) | BWTd (&#8595;) | FWTd (&#8595;) |
> > > | ----- | ----- | ----- | ----- |
> > > | ACIL | 0.29 $\pm$ 0.00 | -0.00 $\pm$ 0.00 | 0.30 $\pm$ 0.00 |
> > > | F-OAL | 0.28 $\pm$ 0.00 | 0.01 $\pm$ 0.00 | 0.28 $\pm$ 0.00 |
> > > | R-MDN | 0.02 $\pm$ 0.01 | -0.00 $\pm$ 0.01 | 0.02 $\pm$ 0.00 |
> > >
> > > As can be seen, both ACIL and F-OAL have excellent BWTd, which means that they effectively mitigate catastrophic forgetting, as their papers propose. However, they result in significantly worse ACCd and FWTd, which means that they make use of confounder information to make predictions (exhibiting large deviations from the theoretical maximum accuracy). Here, R-MDN has better BWTd, ACCd, and FWTd, meaning that it learns confounder-free features for making predictions, thus approaching the theoretical accuracy.
> > >
> > > ---
> > >
> > > > If you deal with medical data only, the title and abstract are too general. It is normal that it is treated as a regular continual learning paper and asked to compared in SOTA fashion.
> > >
> > > The term "medical data" here is quite broad. Yes, we test our models on MRI neuroimaging data from ABCD and ADNI datasets, but we also test on RGB images from the HAM10K dataset, a fully different type of imaging. Additionally, none of our synthetic datasets are medical. When formulating our method, we never impose any constraints that prohibit using R-MDN with models that will be trained on natural images. R-MDN is purely a normalization layer, so its usage is independent of the dataset. Please note that we selected some real-world continual datasets to show how removing confounders is important for many applications, while we also show results on more classic image and synthetic datasets too. If the reviewer thinks it appropriate to add the terms “in medical studies” to the title of this paper, we would be happy to do so.
> > >
> > > Moreover, since our paper focuses not on **task-specific** learning within continual learning but advancing **confounder-free feature learning**, we compare our method to every previously proposed state-of-the-art method such as MDN, P-MDN, and BR-Net. All of these works use similar datasets, thus allowing us to make meaningful comparisons.
> > >
> > > ---
> > >
> > > We hope that this additional discussion and results clarify the reviewer's concerns around the novelty of the method proposed for confounder-free learning, and especially its difference from purely continual learning algorithms like F-OAL.

---

### Official Review · Reviewer_m5AR · 2025-03-12

**Overall Recommendation:** 3

**Summary:**

This paper studies how to remove confounder in continual learning process, and proposes the Recursive-MDN (R-MDN) layer. The R-MDN adopts  statistical regression via the recursive least squares for maintaining an internal state. By removing the confounder factors, the model will be less fitted to the irrelevant information in each task, and therefore the forgetting issue can be mitigated.

The proposed method is integrated with multiple baselines and evaluated over both synthetic and real datasets.

**Claims And Evidence:**

Yes

**Essential References Not Discussed:**

N/A

**Experimental Designs Or Analyses:**

Yes.

**Methods And Evaluation Criteria:**

Yes

**Other Comments Or Suggestions:**

Discuss why baselines proposed in recent years (2023, 2024) are not included in experiments.

**Other Strengths And Weaknesses:**

Strengths:

1. The studied problem is relevant to different machine learning research

2. The experimental results are comprtehensive.

Weakness:

1. No theoretical analysis is provided to justify the proposed method.

2. The performance improvement of the proposed method is not always significant.

3. The adopted baselines are pretty old, and recent baselines are not included.

**Questions For Authors:**

The proposed method is claimed to be able to applied with any deep learning architectures. However, the studied models are mostly for learning over Euclidean data, e.g. images. Therefore, a natural question is: whether the proposed method can be applied to structured data and integrated with models like graph neural networks? Then the setting would be continual graph learning, e.g. 'Online Continual Graph Learning', 'CGLB: Benchmark Tasks for Continual Graph Learning'. I would recommend the authors to discuss this to justify the applicability of the proposed technique.

**Relation To Broader Scientific Literature:**

This paper studies how to boost the performance of continual learning models by removing the confounder factors. From the continual learning perspective, this research would be relevant to different areas, e.g. the current LLMs also require the continual pre-training techniques for keeping an up-to-date knowledge base. From the confounder removal perspective, the contirbution is also likely to be beneficial to other machine learning areas.

**Theoretical Claims:**

There is no theoretical analysis in this work.

---

> ### Author Rebuttal · Authors · 2025-04-01
>
> Thank you for your thorough review and thoughtful feedback—we respond below to the concerns you have raised.
>
> ---
>
> > No theoretical analysis is provided to justify the proposed method.
>
> A more extensive exploration of the theoretical framework would surely be valuable. However, we would like to highlight that our work follows the definitions of MDN, which provides a closed-form solution to the generalized linear model removing the effect of the confounders (this is a common practice in statistics for controlling for confounders). Since the closed-form solution cannot be applied to newer architectures or continual learning settings (see the section on Related Works) because of its dependence on batch statistics and knowing all confounding values (even for future time points), we proposed a recursive reformulation of MDN. Theoretically, these two formulations are equivalent under specific conditions that we mention in Section 3.1 and Suppl. A. Additionally, we explore synthetic setups based on carefully controlled environments for various variables to identify theoretical maximum accuracies achievable that methods can be validated against. As an addition, we are now constructing a second, slightly more complex, synthetic setup to further justify the effectiveness of the proposed method---specifically, we vary both the position as well as the intensity of the confounding variable over various stages of continual learning. Please refer to our response to Reviewer Shhj above (first paragraph) for more details about the setup and results. Results are presented in Figure 1 and Table 2 at https://docs.google.com/document/d/1VPSuH5B_XvGlGoSlg_jCikB8OPLfyI65oSgACuWxzco/edit?usp=sharing, which we will add to the final paper. We hope that this addition can further support our findings.
>
> ---
>
> > The performance improvement of the proposed method is not always significant.
>
> Please see our response to Reviewer s7kX (first paragraph) about this point. Across multiple experiments, we observe that our method achieves substantial improvements over prior works when results are analyzed over multiple different metrics, since baseline models might perform well on specific individual metrics.
>
> ---
>
> > The adopted baselines are pretty old, and recent baselines are not included. Discuss why baselines proposed in recent years (2023, 2024) are not included in experiments.
>
> Baselines for our work are of two different kinds: one that introduces methods for continual learning and two that introduces methods for confounder-free learning. Supervised continual learning is broadly partitioned into three categories: replay-based methods, regularization-based methods, and architecture-based methods. We include a baseline from each of these categories. While this might be non-exhaustive, these methods do not enforce any constraints on learning confounder-free features, so we think that any other recent baseline will perform poorly on our chosen metrics, specifically dcor$^2$. On the other hand, we test our method against the current state-of-the-art confounder-free learning methods such as BR-Net, MDN, and P-MDN, which encompass adversarial-, statistical regression-based, and gradient-based learning. We hope that this provides a comprehensive picture of the landscape.
>
> ---
>
> > …whether the proposed method can be applied to structured data and integrated with models like graph neural networks?
>
> While the focus of our work in this paper has been on learning over Euclidean data (e.g., images), there are no reasons why the proposed method could not be applied to structured data. We used images because the prior work on confounder-free learning that we discussed in the paper also deals with image input, so this allowed for effective comparison. If the input data is of tabular or structured graph forms, as opposed to images, similar layers of neural network (e.g., fully connected for tabular data or graph convolution for graphs) could be applied to the input. This results in mid-level feature representations. Hence, the R-MDN (and all other prior work) could be simply applied on top of the feature representations for de-confounding.
>
> It is noteworthy that online continual graph learning is an emerging direction (Donghi et al., 2025). This is particularly interesting because we can construct causal graphs that capture the interactions of various observed variables (confounders, mediators, etc.) and use them for learning bias-free representations. Exploring the application of our proposed method, as well as other methods for confounder-free learning in static or continual graph learning frameworks, is an exciting direction for future work.

---

> > ### Comment · Reviewer_m5AR · 2025-04-03
> >
> > Thanks for the detailed responses from the authors on theoretical analysis, performance improvement, baselines, and potential extension to graph data.

---

### Official Review · Reviewer_Shhj · 2025-03-15

**Overall Recommendation:** 3

**Summary:**

The paper develops a method to debias intermediate representations in a continual learning setting. The idea is to regress the representation on the biased feature and the label. Then, only the residuals after removing the role biased feature are used as the representations. The experiments show the method improves over existing methods.

**Claims And Evidence:**

The claims made are not proved, which is okay because theory with neural networks is difficult.

Experimentally, the method seems to do well across a verity of tasks.

**Essential References Not Discussed:**

The work does not discuss a recent works like https://arxiv.org/pdf/2404.19132

**Experimental Designs Or Analyses:**

The setup of the experiments vary the distributions over time and this does represent a few scenarios in the problem setup in the paper.

Without the theory, the experiments should have explored more synthetic examples, with varying features beyond position, to study the interaction of changing relationships between features and confounders. For example, the positions of the features affected by the confounder can switch positions of the features that are affected by the non-confounder features.

**Methods And Evaluation Criteria:**

Yes.

**Other Comments Or Suggestions:**

It would be useful to understand why P-MDN works better in some settings.

**Other Strengths And Weaknesses:**

- The method is simple and the choices can be validated , as is done in the method.
- The experiments done on real datasets are encouraging.

Main weakness seems to be to understand what is missing with the method. See Questions.

**Questions For Authors:**

1. I am not sure that I fully appreciate the reasons behind the linear de-confounding : "The assumption of an underlying linear relationship between confounders and learned features arises from two key considerations: (1) decisions made by nonlinear models are often challenging to interpret, and (2) sufficiently powerful nonlinear models can extract almost any arbitrary variable from the information present in the features, even if those variables are not explicitly represented."

You build nonlinear models, so point 1 seems moot? Am I missing something? Second, linear regression is limited to not handle interactions between two coordinates in the representations. So any confounder that affects this interactions will not be removed with the proposed method. Is there reason to believe this is unlikely? I'm asking because it's not apriori clear which kinds of confounders can be debiased.

2.  Can the authors intuitively explain the type of relationships between confounder and input (as a thinking tool for representations) that the proposed method handles?

**Relation To Broader Scientific Literature:**

Continual learning is important and shifts such as those induced by confounding (or spurious features) occur in real life (such as the age being correlated with certain diseases). The findings in the paper combine tools for debiasing with tools from continual learning and that improves accuracy on some real world data.

**Theoretical Claims:**

No theoretical claims. It would have been useful to see that type of biases that said method would get rid of but analyzing internal representations of neural networks is not a solved problem, so this can't be held as a big problem of the paper.

Of course this does not mean no analysis is possible. It would have been very useful to see the type of  shifts over the continuum that the method can handle, maybe in a linear model? Does such analysis exist?

---

> ### Author Rebuttal · Authors · 2025-04-01
>
> Thank you for your thorough review and thoughtful feedback—we respond below to the concerns you have raised.
>
> ---
>
> > Without the theory, the experiments should have explored more synthetic examples…
>
> We agree that additional synthetic setups would strengthen the empirical results we observe. We selected the synthetic setup in the paper, which plays with the distribution of the intensity of main effects and the confounder across different training stages, as it is used extensively in prior work such as by Adeli et al. (2020a), Lu et al. (2021), and Vento et al. (2022). But complex setups seem important. We are now constructing a second synthetic setup which explores the influence of both the position of the confounder as well as the intensities of the main effects and the confounder. More specifically, we generate 1024 32$\times$32 images that are implicitly broken down into 16 8$\times$8 grids. The top left and bottom right grids contain Gaussian kernels of intensity $\sigma_A$, denoting the main effects. The confounder is represented by a Gaussian kernel of intensity $\sigma_B$, whose position varies from the bottom left to top right of the image over 4 different training stages. Both $\sigma_A$ and $\sigma_B$ are sampled from the distribution $\mathcal{U}(3, 5)$ for group 1, and $\mathcal{U}(4, 6)$ for group 2. A fair classifier should remain unaffected by confounder information, irrespective of its position within the image. Such a setup also allows us to compute theoretical maximum accuracy achievable for each training stage that we can validate methods against. Results are presented in Figure 1 and Table 2 at https://docs.google.com/document/d/1VPSuH5B_XvGlGoSlg_jCikB8OPLfyI65oSgACuWxzco/edit?usp=sharing. We will include the same in the final paper.
>
> ---
>
> > The work does not discuss recent works like https://arxiv.org/pdf/2404.19132
>
> Unsupervised continual learning is an interesting framework to work in. We apologize for not discussing related works in the paper, but rest assured we will cite them in the final paper. In our work, we focus on supervised continual learning, where we train classification networks on labeled data. In UCL, the primary challenge is the lack of labeled confounders. Such a scenario is similar to when certain factors that bias learning are unobserved or hidden. Effectively ensuring plasticity, stability, and cross-task consolidation through the objective function, while making sure that intermediate features can be normalized by removing the influence of confounders (after identification) is an exciting opportunity for future work. In medical contexts, labeled information is often present as metadata with collected samples. Thus, techniques from UCL can be coupled with R-MDN to learn confounder-free representations that additionally exhibit minimized forgetting.
>
> ---
>
> > It would be useful to understand why P-MDN works better in some settings.
>
> P-MDN minimizes a proxy to the bi-level optimization problem that MDN tries to minimize with a closed-form solution. P-MDN replaces the MDN’s closed-form solution, which requires a batch-level operation (not friendly for many architects and deep learning), with a proxy objective. Successful minimization of the P-MDN proxy objective depends on the careful tuning of the hyperparameter that controls the loss function computed over the features and the metadata. The success of P-MDN is then affected by this choice of hyperparameter as well as how close the proxy objective represents the true objective to be minimized. In our experiments, we extensively tuned this hyperparameter and analyzed the influence from the learning rate and optimizer and found that gradient updates due to this proxy objective led to large variance in performance across runs (different model seeds), showing improvements in some settings and weaker performance in others. In contrast, our approach is better since it is a recursive reformulation of MDN that theoretically approaches the solution from MDN under specific conditions we analyze in Section 3.1.
>
> ---
>
> > I am not sure that I fully appreciate the reasons behind the linear de-confounding… Can the authors intuitively explain the type of relationships between confounder and input (as a thinking tool for representations) that the proposed method handles?
>
> Similar to the original MDN formulation, R-MDN does not assume that confounders have to linearly influence the input image, because the method does not directly operate on images, but on the feature embeddings, which are non-linear abstractions of the input. Moreover, R-MDN can be applied to feature embeddings at multiple layers, so overall it can effectively remove non-linear confounding between the input and the confounders. However, we admit that neither R-MDN nor MDN has a theoretical guarantee to remove all the non-linear confounding relationships. We will make this discussion more explicit in the final paper.

---

### Official Review · Reviewer_s7kX · 2025-03-21

**Overall Recommendation:** 3

**Summary:**

To remove the influence of confounding variables from intermediate feature representations, the authors introduce the Recursive MDN (R-MDN) layer. They note that such layer can be integrated into any deep learning architecture--including vision transformers--and at any model stage.

R-MDN performs statistical regression via the recursive least squares algorithm to maintain and continually update an internal model state with respect to changing distributions of data and confounding variables.

In the empirical evaluation the authors claim that R-MDN promotes equitable predictions across population groups, for static learning and for different stages of continual learning, by reducing catastrophic forgetting caused by confounder effects changing over time.

**Claims And Evidence:**

Claims

- The authors note that the introduced R-MDN layer can be integrated into any deep learning architecture--including vision transformers--and at any model stage.

- The authors claim that R-MDN promotes equitable predictions across population groups, for static learning and for different stages of continual learning, by reducing catastrophic forgetting caused by confounder effects changing over time.

Evidence

- The authors claim that R-MDN layer can be integrated into any deep learning architecture, only on a small subset of architectures are this claim is evaluated.

- The claim that R-MDN promotes equitable predictions across population groups, for static learning and for different stages of continual learning, by reducing catastrophic forgetting caused by confounder effects changing over time is evaluated on synthetic and real world datasets.

**Essential References Not Discussed:**

I was not able to notice, although, I'm not an expert in the field.

**Experimental Designs Or Analyses:**

The experimental design looks good.

In the experimental design they follow past works and use also exiting datasets from in this domain.

They evaluate on synthetic datasets and publicly available ones.

**Methods And Evaluation Criteria:**

Methods

- The methodology seems valid and clearly explained

Evaluation Criteria

- They define the Average Accuracy distance (ACCd), Backward Transfer distance (BWTd), and Forward Transfer distance (FWTd) metrics. As noted these metrics are adapted from ACC, BWT, and FWT defined by (Lopez-Paz & Ranzato, 2017) to work with the setting where a model is expected to achieve certain theoretical accuracies on data from both previous and future stages of training.

**Other Comments Or Suggestions:**

/

**Other Strengths And Weaknesses:**

Strengths

- Very well introduced, written and nicely structured paper.
- The methodology seems novel and it is clearly explained.
- Empirical evaluations and results on synthetic and publicly available dataset, and comparison with existing work under different continual learning regimes.

Weaknesses
- The claims somewhat not supported by the evaluation, marginal improvements or I was not able to see clear improvement separation
- In the evaluation for the synthetic dataset they use CNN architecture, while in the evaluation on the publicly available dataset they use ViT architecture. I would have expected use of the methodology at 2 or three different NN architectures (to support the claim in the abstract and add more value to the paper).
- On the metrics that they introduce it looks like the proposed approach does not have favorable scores.

**Questions For Authors:**

I have seen that the proposed layer was applied and placed at different layers, Have the authors checked the performance over different NN architectures under one setup to check what kind of improvement we can get?

**Relation To Broader Scientific Literature:**

The authors consider the residual r from the expression z = x˜β˜x + yβ˜y + r = Xβ + r, where X = [x y ˜ ] and β = [β˜x; β˜y] is a set of learnable parameters. They take learned features z that are first projected onto the subspace spanned by the confounding variable and the labels, with the term x˜β˜x corresponding to the component in z explained by the confounder, and yβ˜y to that explained by the labels.

Practically they compute the residual r = z − x˜β˜x; i.e., only with respect to β˜x. They focus on this residual that explains the components in the intermediate features irrelevant to the confounder which seems relevant to the labels and, thus, for the classification task.

**Theoretical Claims:**

I have not cheeked the theoretical claims, the derivations seem right (related to equations 2, 3, and 4) but have not checked the detailed in the Appendix.

---

> ### Author Rebuttal · Authors · 2025-04-01
>
> Thank you for your thorough review and thoughtful feedback - we respond below to the concerns you have raised.
>
> ---
>
> > … marginal improvements or I was not able to see clear improvement separation
>
> In this work, we introduce a method that allows DNNs to learn confounder-free features during continual learning. A consequence is that the model promotes equitable predictions across class categories. Thus, we must analyze results across multiple metrics, as a baseline model might perform well on specific individual metrics.
>
> For example, in Exp. 4.1, a baseline CNN scores the best on the BWTd metric, but comparably on ACCd and FWTd. Here, lower is better all three metrics. Similarly, a stage-specific MDN model does poorly on ACCd. P-MDN, on the contrary, does well on ACCd and FWTd, but not on BWTd. With our approach, we observe that the method achieves competitive performance across all metrics, significantly outperforming other methods in metrics they perform poorly at (such as 1% improvement on FWTd but 10% on ACCd when compared to stage-specific MDN). In Exp. 4.3, while our approach leads to a modest 2% improvement on the dcor^2 (CN) metric, it has a significant 15% improvement on dcor^2 (AD) when compared to P-MDN. P-MDN is biased towards the AD category, while our approach has similar ranges of dcor^2 across all categories. Furthermore, significant improvements can be seen in Figure 3 (as evidenced by the five straight lines) and Figure 6 (as evidenced by the identification of the cerebellum).
>
> We believe that these, along with other results from the paper, provide clear evidence of substantial improvement over prior works for the claims we make, as analyzed through a combination of qualitative and quantitative evaluations.
>
> ---
>
> > I would have expected use of the methodology at 2 or three different NN architectures …
>
> Please note that we tested our method on both CNN and ViT with a mix of datasets and different architectural setups; though, we agree that evaluating the method on more DNN architectures would add more value to the paper. We selected a CNN and ViT model backbone because they lend themselves well to the task of image categorization, which is what we focus on in this paper. Synthetic datasets (as presented in Exp. 4.1 and Suppl. C) are evaluated using a CNN backbone to enable direct comparison to prior de-biasing methods in this space such as MDN, P-MDN, and BR-Net, all of which adopt a similar backbone and model size. This helps us understand how well our method performs. Additionally, we include an experiment on the ABCD dataset (Exp. 4.4) using a 3D CNN model to extend its application to a real-world setup. Finally, we evaluate a 2D ViT (Exp. 4.2) and a 3D ViT (Exp. 4.3) on different real-world setups to analyze their performance. To the best of our knowledge, this is the first time transformers have been used for confounder-free feature learning in this context. We hope that the demonstration of both 2D and 3D CNNs and ViTs will add value to our claim. To complete the loop and further strengthen our claim, we are now running an experiment with a ViT as a model backbone on the synthetic dataset. Results are presented in Table 1 at https://docs.google.com/document/d/1VPSuH5B_XvGlGoSlg_jCikB8OPLfyI65oSgACuWxzco/edit?usp=sharing.
>
> ---
>
> > On the metrics that they introduce it looks like the proposed approach does not have favorable scores.
>
> Across the different metrics, our method favorably improves scores on dcor^2, which quantifies the influence of the confounder on learned features. Improvement is seen through a decrease in  dcor^2 in Tables 2, 3, and 4, often across all class categories to reduce inequitable predictions. On the accuracy metric, scores sometimes modestly drop to reflect the model not learning using shortcuts by looking at confounder information for prediction. Only for Exp. 4.2 we see that our approach leads to a slightly weaker score on FWT, possibly due to the corresponding influence from a slightly higher accuracy due to utilizing confounder information, which we seek to avoid. Since we can design carefully controlled environments through synthetic datasets, we can analyze the true theoretical accuracy, BWT, and FWT a fair model should get, and we see favorable scores across each in Exp. 4.1.
>
> ---
>
> > Have the authors checked the performance over different NN architectures under one setup…
>
> Yes, the placement of the R-MDN layer is an important hyperparameter that must be tuned to the specific DNN architecture being used. We analyze various placements across different architectures through Exp. 4.2 and Suppl. H. Prior work (Lu et al., 2021) has shown that applying MDN after both convolutional and fully connected layers results in the best performance in CNNs. We see similar performance improvement in CNNs with R-MDN in Suppl. H. We construct a similar setup for ViTs in Exp. 4.2 and find that adding R-MDN only after the pre-logits layer provides the best performance.

---

### Decision · Program_Chairs · 2025-05-01

**Decision:**

Accept (poster)

**Comment:**

This paper proposes Recursive Metadata Normalization (R-MDN), a novel and principled approach to removing confounding influences from intermediate feature representations in continual learning. Built upon recursive least squares (RLS), R-MDN operates online and can be flexibly inserted into deep networks, including vision transformers, without requiring batch-level statistics or task-specific architectures. The authors demonstrate its efficacy across synthetic and real-world datasets (e.g., HAM10000, ADNI, ABCD), showing improved fairness, backward and forward transfer, and confounder invariance.

While there has been some opinion that proposed method only gives marginal improvements (s7kX), lacks theoretical analysis (m5AR), the reviewers agreed that the flexibility of R-MDN, which can be plugged in any deep learning models, and positive experimental results on multiple datasets are worth publishing. Hence, the decision is Accept. But, AC strongly recommends the authors to incorporate the following crucial discussions / additional results in the final version:

  - Additional results for ViT architecture on synthetic data.
  - More clear discussion on linear confounding.
  - Discussions and comparison with P-MDN.
  - Clear definition on the metric $dcor^2$.